# Birds of a Feather, Flocked Together? Deriving Background-Invariant Classifiers from CLIP Image Backbones

## Abstract

CLIP image encoders are widely used for image classification, yet recent work indicates that they remain vulnerable to systematic biases that undermine robustness. In particular, correlations between foreground objects and their backgrounds constitute a salient and practically important class of spurious dependencies. In this paper, we systematically quantify the susceptibility of CLIP image encoders to such background-based spurious correlations. We identify characteristic trends in both the image and text encoders that help explain this behavior and develop a new mitigation method. Our method is the first approach to achieve worst-group accuracy exceeding 90% on Waterbirds under perfect spurious correlation (no minority-group examples in the training data). It demonstrates state-of-the-art performance across benchmarks and is highly practical, relying exclusively on synthetic data with strong sim-to-real transfer.

## 1. Introduction

CLIP image encoders have become widely adopted backbones for image classification, both in zero-shot settings (Radford et al., 2021) and as frozen feature extractors paired with lightweight heads or adapters (Gao et al., 2024). Their flexibility and strong transfer performance have led to wide spread adoption in downstream tasks such as recognition, detection, retrieval, and multimodal reasoning (Khan et al., 2022; Zhang et al., 2024).

However, CLIP encoders exhibit considerable levels of bias and sensitivity to spurious signals (Maheronnaghsh & Alvanagh, 2024; Wang et al., 2024; Varma et al., 2024). Specifically, CLIP's performance degrades in low-income or non-Western contexts (Nwatu et al., 2023; Pouget et al., 2024)

[1]Anonymous Institution, Anonymous City, Anonymous Region, Anonymous Country. Correspondence to: Anonymous Author <anon.email@domain.com>.

Preliminary work. Under review by the International Conference on Machine Learning (ICML). Do not distribute.

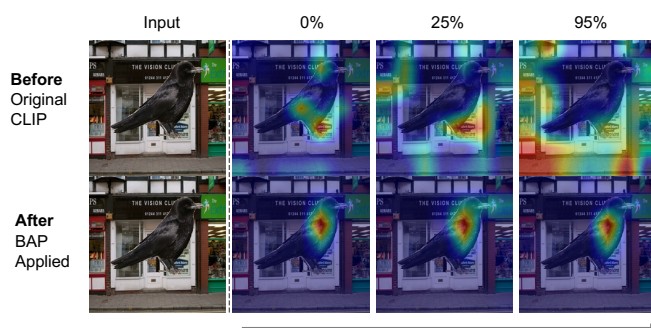

*Figure 1.* **Visualizing Background Invariance via Grad-CAM (Selvaraju et al., 2019).** As the spurious correlation ratio in the training data increases (left to right), standard CLIP-pretrained encoders significantly shift their focus toward environmental cues. In contrast, our BAP method successfully anchors the model's attention to the foreground object (bird) and maintains uniform focus on core features regardless of the prevalence of the spurious signal.

and on tailored spurious-feature benchmarks like CounterAnimal, where it can even underperform standard supervised models (Wang et al., 2024).

A particular source of spurious errors, which has significant practical downsides, is background spurious correlations. Backgrounds can often encode strong but coincidental links with foreground objects. For example, waterbirds may often appear over water backgrounds while landbirds appear against land backgrounds (Sagawa et al., 2020). Of high practical concern are spurious background correlations in medical imaging, where certain diseases may be misclassified due to erroneous links to specific scanners or acquisition protocols (Vasquez-Venegas et al., 2025; Dehkharghanian et al., 2023).

CLIP-based classifiers may be specifically prone to background bias since CLIP is trained contrastively to align images with natural language descriptions that often mention both objects and their typical contexts (Lin et al., 2024; Anonymous, 2025). This training regime encourages the image backbone to encode correlations between foreground and background signals. This causes downstream linear heads to inherit dependencies on background patterns, which can hurt generalizability and cross-domain stability

(Wang et al., 2024; Agarwal et al., 2025).

In this work, we explain CLIP's background bias by showing its encoders act as compositional "bag-of-features" models, representing scenes as a linear superposition of foreground and background rather than deep contextual transformers. Building on this insight, we introduce Background-invariant Anchor Pre-training (BAP), a practical approach to robustify visual backbones before they are exposed to target downstream distributions. BAP utilizes a two-phase protocol that leverages synthetic data. Our method achieves state-of-the-art results, including a worst-group accuracy exceeding 90% on Waterbirds, even under the traditionally challenging setting of perfect spurious correlation. Furthermore, we demonstrate that BAP recasts robustness as a task-agnostic pre-training stage, facilitating strong sim-to-real transfer and robust background invariance even for objects omitted during the pre-training phase. Our primary contributions are summarized as follows:

- We provide a formal account of CLIP's background bias by quantifying "embedding additivity," demonstrating that both image and text encoders represent scenes as a linear superposition of constituent parts.

- We introduce an annotation-free, two-stage pre-training method that robustifies CLIP image encoders by aligning them with foreground-only anchor vectors to produce background-invariant representations.

- We achieve a state-of-the-art worst-group accuracy (WGA) exceeding 90% on Waterbirds under 100% spurious correlation, a setting where both training and validation sets lack minority-group examples.

- We demonstrate that BAP-induced robustness generalizes to the super-class level. Once pre-trained, the same feature extractor can be reused for multiple downstream tasks using standard Empirical Risk Minimization (ERM), maintaining stability regardless of the spurious correlation strength in the downstream data and achieving robust OOD performance.

The code to reproduce our results is available at our GitHub repository.

## 2. Related Work

Many approaches have been developed to mitigate spurious correlations when group annotations are available. Among those, Deep Feature Reweighting (DFR) (Kirichenko et al., 2022) has established itself as a strong oracle baseline. Alongside DFR, group-robust training regimes such as Group Distributionally Robust Optimization (Group DRO) (Sagawa et al., 2020) directly optimize worst-group performance during training. Other oracle methods, such as Progressive Data Expansion (PDE) (Deng et al., 2023), use

group labels to construct a training curriculum that starts from group-balanced data and progressively incorporates more imbalanced samples. However, since DFR attains stronger performance compared to Group DRO and PDE, we use DFR as the representative oracle baseline in this work.

To relax the requirement of group annotations, methods such as Just Train Twice (JTT), SELF, and Automatic Feature Reweighting (AFR) (Liu et al., 2021; LaBonte et al., 2023; Qiu et al., 2023) aim to automate the construction of balanced training or validation sets by identifying challenging examples using model-based signals such as loss or training difficulty. AFR achieves state-of-the-art results in this category; we therefore select it as a representative non-oracle last-layer retraining baseline.

A limitation of the above approaches is their reliance on the presence of minority groups in the training set. Consequently, these methods fail in "extreme" cases where the spurious feature has a 100% correlation rate with the class label (Liu et al., 2025). A further limitation is that these methods focus on example reweighting or last-layer adjustment, as opposed to modifying the feature extractor's representations. As a consequence, the embedding space continues to encode both core and spurious features (Kirichenko et al., 2022; LaBonte et al., 2023), which relegates robustness to a post-hoc tuning task. This hampers transferability to settings where only the representation, but not the reweighting procedure, is reused (Xu & Jaakkola, 2021; Burns & Steinhardt, 2021).

Beyond reweighting, representation-level methods like PruSC (Le et al., 2024) use geometric or pruning-based objectives to extract "spurious-free" sub-networks without group labels. While these methods introduce complexity, such as clustering or contrastive retraining, they act directly on the backbone to produce a spurious-free representation. PruSC in particular, achieves relatively high worst-group accuracies (75%–90%). Therefore, we include PruSC as a state-of-the-art representation-level baseline, complementing reweighting methods (DFR, AFR) to provide a representative envelope of spurious correlation mitigation baselines.

Finally, since our primary interest is in CLIP-based models, we specifically adopt WiseFT and RoboShot (Wortsman et al., 2022; Adila et al., 2023) as strong CLIP-oriented robustness baselines. RoboShot generates spurious feature insights (using LLMs) which are used to adapt CLIP's representations and decision boundaries through embedding-space operations while WiseFT achieves robustness through weight-space ensembling of zero-shot and fine-tuned models.

## 3. Quantifying Rare Spurious Correlations

While the Waterbirds benchmark (Sagawa et al., 2020) relies on a fixed, 95% correlation rate between background types and class labels, it remains unclear how CLIP robustness degrades when spurious cues are present yet *rare*. To investigate this, we propose a flexible adaptation of the CUB-Places paradigm ( (Wah et al., 2011; Zhou et al., 2017)). Unlike standard binary setups, we partition backgrounds into three disjoint functional sets:

**1. Neutral Set** $B$: containing semantically disjoint scenes to serve as a control. We selected *Rainforests* for this set. Since $B$ appears for both waterbird and landbird classes with equal probability, we ensure that the mutual information between this set and the class label is minimized.

**2. Waterbirds Spurious Set** $B_W$: associated with Waterbirds. We selected *Snowfields* for this set.

**3. Landbirds Spurious Set** $B_L$: associated with Landbirds. We selected *Shopfronts* for this set.

We departed from the more standard bird-background pairings used by Waterbirds to produce a selection of background types that had the least possible semantic overlap.

We define the training distribution via a tunable spurious rate $\alpha$. For a given class $Y$ associated with spurious background $B_{target}$, training samples are drawn from $B_{target}$ with probability $\alpha$ and from $B$ with probability $1 - \alpha$. This allows us to sweep the spurious correlation from non-existent ($\alpha = 0$) to the *rare spurious regime* ($\alpha \leq 0.1$) and all the way to more frequent spurious backgrounds ($\alpha \geq 0.5$). For example, at $\alpha = 0.1$, the training set for Waterbirds consists of 90% samples on neutral backgrounds and 10% on Snowfields, mirrored symmetrically for Landbirds (90% neutral, 10% Shopfronts).

We evaluate performance using two metrics: **Neutral Accuracy**, measured on held-out foregrounds superimposed on neutral backgrounds $B$ to assess core feature learning; and **Trap Error(s)**, measured on counterfactual foreground-background pairings (e.g., Waterbirds on $B_W$).

### 3.1. Preliminary Observations

As illustrated in Figure 2, we observe distinct failure modes in CLIP architectures:

**(1) Higher Susceptibility:** CLIP image encoders exhibit significantly higher Trap Errors than their ImageNet-supervised counterparts. This gap is most pronounced in the ConvNeXt architecture (Liu et al., 2022), where the CLIP variant yields nearly $4\times$ the error of the supervised variant at $\alpha = 0.5$. **(2) Sensitivity to Rare Cues:** CLIP encoders (and to a lesser extent, supervised ViTs) retain high spurious reliance even when correlations are rare. At $\alpha = 0.1$, where spurious

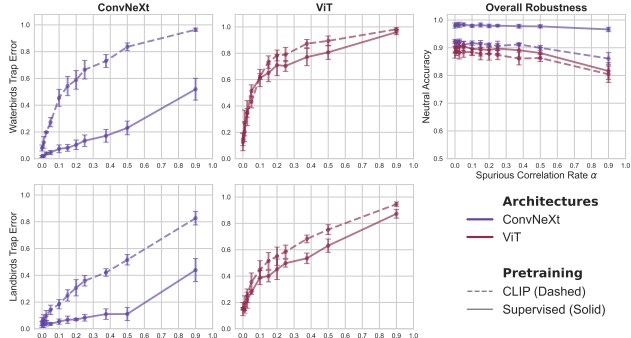

*Figure 2.* **Impact of Spurious Correlation Rate ($\alpha$) on Model Error and Robustness.** The left and center columns display the spurious "Trap Error" (lower is better) for waterbird and landbird classification across ConvNeXt and ViT architectures. The rightmost column reports Neutral Accuracy on the neutral background test set. Results compare CLIP pre-training (dashed lines) against Supervised pre-training (solid lines). Error bars represent standard deviation across 5 seeds.

backgrounds appear in only 10% of training data, CLIP ViT (Dosovitskiy et al., 2021) and ConvNeXt models approach a 50% trap error rate for Waterbirds. **(3) Feature Co-learning:** While overall robustness (Neutral Accuracy) degrades slightly with higher $\alpha$, it does not drop as precipitously as the Trap Error climbs. This supports the observation that large-scale pre-trained backbones learn spurious and core features simultaneously rather than trading one for the other (Kirichenko et al., 2022).

## 4. Understanding CLIP's High Background Bias

In order to understand CLIP models' tendency for higher levels of background bias relative to their supervised counterparts, we test whether scene representations behave as a linear superposition of separate object and background components. For each paired object image $I_a$, background image $I_b$, and composite scene $I_{a,b}$, an encoder $g(\cdot)$ produces unit-norm embeddings:

$$v_a = g(I_a), \quad v_b = g(I_b), \quad v_{a,b} = g(I_{a,b}).$$

We quantify an image encoders tendency to represent backgrounds and foregrounds independently by measuring the cosine similarity between the composite embedding and the vector sum of its parts:

$$\text{Sim}_{\text{Both,Sum}}(I_a, I_b) = \cos(v_{a,b}, \, v_a + v_b). \quad (1)$$

A High $\text{Sim}_{\text{Both,Sum}}$ score indicates that the scene is encoded as a near-linear blend of independently represented object and background directions. We apply Equation (4) to the embeddings of 3000 random pairings of CUB birds and Places365 backgrounds and present the results in Table 1.

*Table 1.* Mean cosine similarities ($\pm$ standard deviation) across 3000 foreground–background image pairs. Higher $\text{Sim}_{\text{Both,Sum}}$ values for CLIP mirror the robustness gaps observed in Figure 2.

| Pretraining | ViT-B/16 | | ConvNeXt-W | |
| --- | --- | --- | --- | --- |
| | Supervised | CLIP | Supervised | CLIP |
| $\text{Sim}_{\text{Both,Sum}}$ | $0.74_{\pm 0.02}$ | $0.85_{\pm 0.04}$ | $0.69_{\pm 0.03}$ | $0.83_{\pm 0.05}$ |

The observations in Table 1 help shed light on the results in Section 3.1: CLIP models exhibit consistently higher $\text{Sim}_{\text{Both,Sum}}$ scores (0.85 for ViT-B/16 and 0.83 for ConvNeXt-W) than their supervised ImageNet counterparts (0.74 and 0.69, respectively). These results support a compositional model where CLIP represents foregrounds and backgrounds as nearly independent directions; consequently, background components remain strongly encoded in the embedding space, amplifying background-based spurious errors. Additionally, we find that the supervised ConvNeXt model, the most robust as per Figure 2, has the lowest $\text{Sim}_{\text{Both,Sum}}$ score, while the architecture with the largest score gap (ConvNeXt) shows the highest discrepancy in robustness.

Our findings are further replicated in CLIP text encoders, which behave similar to 'bag-of-words' (Csurka et al., 2004) models such as Word2Vec (Mikolov et al., 2013) rather than contextual transformer models like BERT (Devlin et al., 2019). This indicates a consistent mechanistic behavior across both modalities; full quantitative results for the text-encoder analysis are provided in Appendix A.

The near-linear separability of foreground and background representations within the CLIP embedding space presents an opportunity for us to recover isolated foreground signals from full scene representations. Our method is designed to take full advantage of this property.

## 5. Methodology

We begin by reformulating the problem of mitigating spurious correlations not as a constraint during downstream training, as with DFR, SELF, and AFR (Kirichenko et al., 2022; LaBonte et al., 2023; Qiu et al., 2023), but as a *robustness pre-training* stage. The core objective of our method, Background-invariant Anchor Pre-Training (BAP), is to condition the visual backbone before it is ever exposed to the spuriously correlated target distribution. Our aim is to yield a feature extractor that is inherently resistant to the background spurious correlations prevalent in datasets like Waterbirds. Our formulation of inducing robustness as a pre-training stage has two distinct advantages over existing methods:

1. **Task-Agnostic Robustness.** Unlike methods such as DFR and PruSC (Kirichenko et al., 2022; Le et al., 2024) which require re-optimization for each downstream task, BAP yields a general-purpose, background-invariant backbone. This allows deployment across varied tasks using standard ERM, without specialized robustness constraints during linear probe fine-tuning.

2. **Resilience to Correlation Strength.** By inducing robustness prior to downstream exposure to the spurious dataset, BAP remains stable regardless of the spurious correlation severity in the target distribution. Performance does not degrade as the prevalence of the spurious signal increases in the downstream target set.

### 5.1. The BAP Algorithm

BAP is composed of two distinct, sequential phases, see Figure 3. The first involves extracting a set of foreground-only anchor vectors in the image encoder's embedding space using a frozen, pre-trained teacher. The second step involves fine-tuning the student model to map the object, appearing in novel, randomized contexts, to this invariant anchor vector.

#### 5.1.1. PHASE 1: ANCHOR VECTOR EXTRACTION

We begin by constructing a dictionary of foreground-only anchor vectors (training target vectors) for the foreground objects used during the robustness pre-training stage (each foreground object gets its own anchor vector). We utilize a frozen, contrastively pre-trained CLIP image encoder, $f_\theta^*$, as a teacher model to generate these anchors.

For a specific foreground object instance ($x_{fg}$) we generate a set of composites by superimposing the object onto diverse, randomly sampled background images, $b_k$. We compute the normalized feature embedding, ($\bar{z}$) for each composite and derive the anchor ($\mathbf{a}$) by averaging these vectors and normalizing:

$$\bar{z} = \frac{1}{K} \sum_{k=1}^{K} f_\theta^*(\mathcal{C}(x_{fg}, b_k)), \quad \mathbf{a} = \frac{\bar{z}}{\|\bar{z}\|_2}, \qquad (2)$$

where $\mathcal{C}$ denotes the compositing function.

By averaging embeddings of the same object across unrelated backgrounds, the variance attributable to background noise is suppressed, while the invariant foreground signal is

retained. We use $K = 16$ for all experiments in this paper; for an investigation into the selection of the $K$ value, see Appendix F. See Appendix E for further theoretical motivation behind anchor vector extraction.

### 5.1.2. PHASE 2: ROBUST ALIGNMENT PRE-TRAINING

In the second phase, we unfreeze the visual encoder ($f_\theta$) and optimize it to align its embeddings with the pre-computed anchors. During this phase, each foreground instance is composited onto a number of random backgrounds. The model is then optimized to minimize the cosine distance between the embedding of these randomized inputs ($\hat{x}$) and the foreground-only anchor ($\mathbf{a}$):

$$\mathcal{L}_{align} = 1 - \frac{f_\theta(\hat{x})^\top \mathbf{a}}{\|f_\theta(\hat{x})\|_2 \|\mathbf{a}\|_2} \tag{3}$$

This objective explicitly forces the embedding layer to align with the pre-computed anchors which induces robustness to changing backgrounds. Furthermore, by utilizing the CLIP's original representations as anchors, we preserve the rich semantic structure of the foreground objects learned during contrastive pre-training. For an in-depth investigation into the effects of this objective on the embedding space, see Appendix B.

### 5.2. Evaluation Protocol and Baselines

To rigorously assess the quality of the learned features, we employ a strict **two-stage evaluation protocol:**

1. Robustness Pre-training: The visual backbone is updated using the generated composites. The two stage pre-training logic that constitutes BAP, Section 5.1, is implemented in this stage.

2. Linear Probing: Following BAPs pretraining, we freeze the visual backbone and train a fresh linear probe on the downstream, spuriously correlated dataset (e.g., Waterbirds) and report the performance on this probe. We use standard cross-entropy on the class labels to train this probe with no robustness-enhancing modifications; the ability to use standard ERM on the spuriously correlated downstream dataset while maintaining high robustness is one of BAPs main practical advantages over existing methods.

Note that freezing the image backbone is crucial in this stage since full fine-tuning using a spuriously correlated dataset causes the encoders robustness to regress to its pre-BAP levels, see Appendix H. We do not anticipate this to be a burden as regards practical deployment since BAP demonstrates strong out-of-distribution (OOD) performance, seen in Sections 6.2 and 6.3.

**Data-Matched ERM Control.** A critical confounder of BAP's pre-training setup is data exposure; simply exposing the model to a large variety of objects on randomized backgrounds might improve robustness regardless of the alignment objective. To isolate the efficacy of BAP's pre-training logic, we introduce a **Data-Matched ERM Control**.

This baseline uses the *exact same* synthetic images and compute budget as BAP. However, instead of aligning to robust anchors, the model is trained via Empirical Risk Minimization (ERM) using standard Cross-Entropy loss on the ground-truth class labels. To ensure this baseline is competitive, we employ a Linear-Probe then Fine-Tune (LP-FT) schedule (Kumar et al., 2022) to preserve pre-trained features better than direct fine-tuning. Following this stage (fine-tuning via ERM), we freeze the visual backbone and train a fresh linear probe on the original, spuriously correlated dataset (e.g., Waterbirds) thus replicating BAPs two stage approach. By holding the training pixels, hyperparameters, compute budget and evaluation protocol constant, any performance gap between BAP and this control can be causally attributed to the anchor alignment methodology.

## 6. Results

We present results on a wide variety of datasets and contexts to demonstrate the universality and generality of our method.

We evaluate BAP across a diverse suite of benchmarks designed to test specific facets of robustness. We use Waterbirds (Sagawa et al., 2020) for our primary results alongside CUB-200-2011 and Places365 to construct our synthetic training data. Furthermore, we utilize CounterAnimal (Wang et al., 2024), MS-COCO 2017 (Lin et al., 2014), and NICO++ (Zhang et al., 2023) to assess OOD generalization across varied domains and real-world contexts.

### 6.1. Waterbirds: Initial Findings

To evaluate our two-stage deployment protocol, we construct robustness pre-training data by composing segmented CUB birds onto randomly sampled Places365 backgrounds. Both the BAP and the Data-Matched ERM Control utilize identical bird-background combinations, dataset sizes, and epoch counts, thereby controlling for data exposure. To prevent data leakage, we restrict our synthetic set to CUB bird instances contained in the original Waterbirds training set.

The second stage utilizes the standard Waterbirds dataset; we use training set for linear probe training and the test set for evaluation. For the Waterbirds-100% experiments, we remove all minority groups (waterbirds-on-land and landbirds-on-water) from the training and validation sets to create a training environment where the class label is perfectly correlated with the background. The test set remains unaltered from the original Waterbirds set for both 95% and 100%

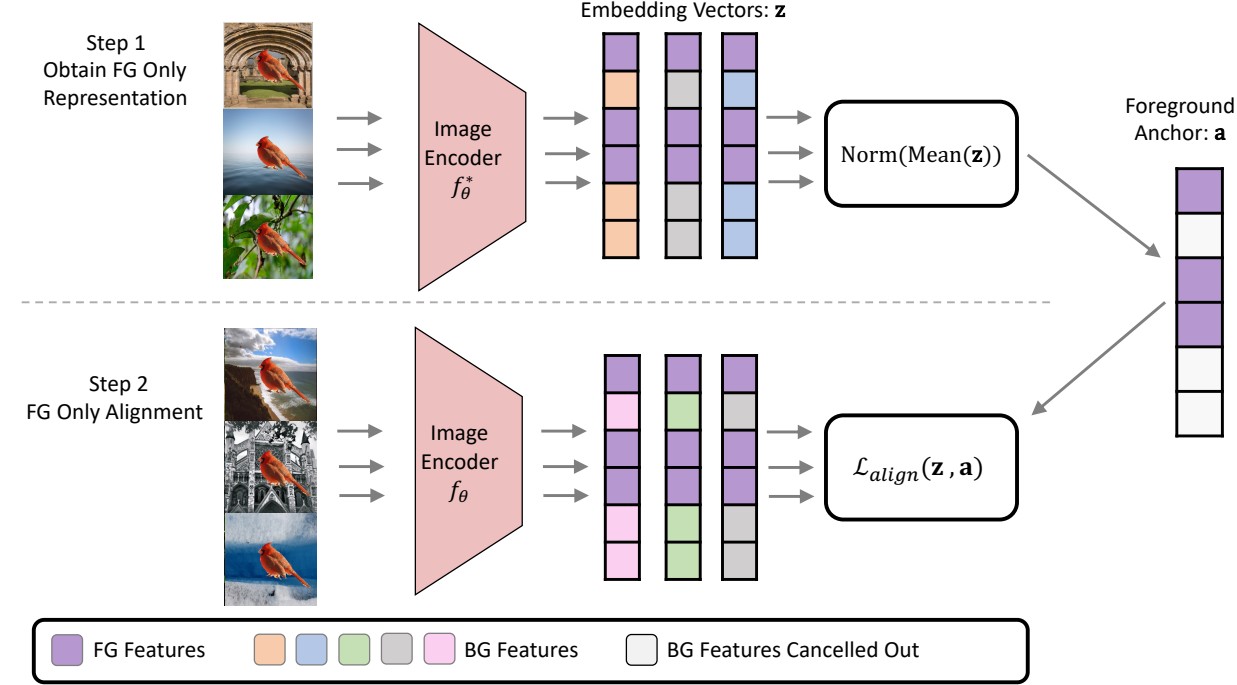

*Figure 3.* **Overview of BAP). (Top)** We generate a robust anchor **a** by averaging embeddings of a fixed foreground object composited onto randomized backgrounds using a frozen teacher encoder $f_\theta^*$. **(Bottom)** The student encoder $f_\theta$ is optimized via $\mathcal{L}_{align}$ to map the object to its foreground-only anchor thus inducing background invariance. Purple slots represent core foreground features, while white slots represent suppressed background features; all other colored slots represent various random background features.

experiments.

To summarize, two ViT-B/16 models were pre-trained on synthetic data, both initialized from LAION-pretraining weights (Schuhmann et al., 2022): one using BAP and the other a Data-Matched ERM Control using standard cross-entropy. Both of these are then fine-tuned as linear probes on the original Waterbirds training set. We compare these against established baselines (e.g., DFR, PruSC, RoboShot) which utilize standard LAION-pretrained encoders and are trained directly on the Waterbirds training set using their respective algorithms. All models are tested on the unaltered Waterbirds test set.

The results in Table 2 demonstrate that BAP consistently outperforms established spurious correlation mitigation techniques for Waterbirds$-95\%$. Notably, BAP slightly exceeds the performance of the DFR oracle ($92.1\%$ vs $91.3\%$ WGA) on Waterbirds-$95\%$. When fine tuning the image backbone as a linear probe, our method yields a nearly **30-percentage-point increase in WGA** over standard LAION-pretrained encoders (Linear Probe baseline). Furthermore, our method achieves nearly $10\%$ higher WGA compared to the data-matched ERM control, indicating that the robustness induced by our method is not exclusively due to increased data exposure.

The gap in WGA grows when investigating the results for Waterbirds-$100\%$. While traditional ERM and Linear Probing suffer a catastrophic drop in WGA (exceeding 35 per-

centage points), **BAP maintains a WGA above** $90\%$ **despite perfect spurious correlation** (to our knowledge, a new state-of-the-art). Furthermore, all other spurious correlation mitigation methods' performance deteriorates significantly when the minority groups are absent from the training set. BAP exhibits nearly identical performance across both $95\%$ and $100\%$ correlated datasets suggesting our pre-training regime facilitates robust background invariance regardless of the strength of spurious correlations in the downstream dataset. Furthermore, while the Data-Matched ERM Control does offer some inherent robustness compared to LAION-pretrained models, it still experiences a $11\%$ drop in WGA when moving from the $95\%$ to the $100\%$ confounded set, further strengthening the claim that increased data exposure is not sufficient to reach the robustness levels exhibited by BAP.

The ability to fine-tune BAP pre-trained models on datasets lacking minority groups without performance degradation offers unique practical advantages over existing methods.

### 6.2. CounterAnimal Results: Sim-to-Real performance

Results from Section 6.1, while promising, do not establish whether robustness gained from synthetic composites carries over to non-synthetic images. To test the performance of BAP in real world settings, we adapt our bird classification task to the CounterAnimal (Wang et al., 2024) benchmark.

*Table 2.* Average and worst-group accuracies (mean $\pm$ std over 5 runs) across various baselines using CLIP ViT-B/16. We group the baselines using the following logic; first are methods that utilize CLIPs native representations, second are post-hoc retraining methods, and third are methods that update the native clip representations in various manners. Note that we omit the last-layer retraining baselines (DFR, AFR) for the 100% benchmarks since these methods rely on data from the minority class and it has already been established (Liu et al., 2025) that they fail in the absence of minority group data.

| Method | Waterbirds-95% (Original) | | Waterbirds-100% | |
|---|---|---|---|---|
| | AVG ($\uparrow$) | WGA ($\uparrow$) | AVG ($\uparrow$) | WGA ($\uparrow$) |
| ZeroShot | $74.5_{\pm 0.0}$ | $51.3_{\pm 0.0}$ | $74.5_{\pm 0.0}$ | $51.3_{\pm 0.0}$ |
| RoboShot | $71.3_{\pm 0.0}$ | $58.0_{\pm 0.0}$ | $71.3_{\pm 0.0}$ | $58.0_{\pm 0.0}$ |
| Linear Probe | $79.8_{\pm 0.6}$ | $61.6_{\pm 0.9}$ | $62.4_{\pm 0.3}$ | $24.5_{\pm 0.9}$ |
| AFR | $91.5_{\pm 0.5}$ | $89.1_{\pm 1.6}$ | - | - |
| DFR | $93.4_{\pm 0.5}$ | $91.3_{\pm 0.6}$ | - | - |
| ERM (LP-FT) | $88.8_{\pm 0.2}$ | $72.6_{\pm 0.4}$ | $62.4_{\pm 0.7}$ | $23.8_{\pm 1.2}$ |
| ERM + RandAugment | $90.6_{\pm 0.8}$ | $73.8_{\pm 3.0}$ | $59.4_{\pm 0.9}$ | $26.7_{\pm 1.2}$ |
| WiSE-FT | $87.5_{\pm 0.1}$ | $67.0_{\pm 0.3}$ | $61.8_{\pm 0.4}$ | $22.0_{\pm 0.3}$ |
| PruSC | $83.3_{\pm 2.8}$ | $76.4_{\pm 3.9}$ | $56.5_{\pm 3.1}$ | $46.9_{\pm 4.2}$ |
| Data-Matched ERM Control | $91.1_{\pm 1.4}$ | $82.7_{\pm 3.6}$ | $86.7_{\pm 2.7}$ | $71.4_{\pm 6.9}$ |
| **BAP** (Ours) | $\mathbf{94.3}_{\pm 0.6}$ | $\mathbf{92.1}_{\pm 1.2}$ | $\mathbf{94.2}_{\pm 0.5}$ | $\mathbf{91.8}_{\pm 0.9}$ |

Unlike Waterbirds' synthetic composites, CounterAnimal consists of natural images containing animals in common contexts (e.g., polar bears on ice) and counter-contexts (e.g., polar bears on grass) which allows us to investigate BAPs sim-to-real performance.

Our pretraining phase retains the same composites of segmented CUB birds on Places365 backgrounds as in Section 6.1 for both BAP and the data-matched ERM control. However, we substitute the downstream classification dataset with real bird images from CounterAnimal which represents a realistic deployment of our method (pre-training on synthetic data, fine-tuning on real data).

We enforce a perfect spurious correlation (100%) when constructing the downstream CounterAnimal training data. For example, in a binary classification task between grouse and prairie-chickens, the training set is constructed such that all grouse appear on snow, while all prairie-chickens appear on grass. Model performance was assessed on the counter-contextual bird-background pairings (in this case grouse on grass and praire-chickens on snow). In addition, we withhold 20% of the training set for testing which allows us to assess model performance across all 4 bird-context pairings. This setup presents a multidimensional challenge since the model is required to overcome both a perfect spurious correlation and a strong sim-to-real distribution shift.

We evaluated the robustness of a single BAP-pre-trained encoder across two distinct downstream binary classification tasks. Specifically, the image backbone remained frozen as a feature extractor, with BAP applied only once during pre-training, followed by independent linear probes for each task (the same logic applies to the data-matched ERM control baseline). The first pairing is between Bramblings and Bulbuls, and the other is between Ptarmigans and Prairie

Chickens (these pairs are moderately challenging, as the birds share taxonomic orders, silhouettes, and camouflage).

*Table 3.* Average and worst-group accuracies (mean $\pm$ std over 5 runs) across various baselines for two pairs of binary classification on the CounterAnimal dataset using ViT-B/16.

| Method | AVG ($\uparrow$) | WGA ($\uparrow$) |
|---|---|---|
| *Pair 1: Brambling vs. Bulbul* | | |
| BAP (Ours) | $\mathbf{92.0}_{\pm 0.7}$ | $\mathbf{86.4}_{\pm 3.3}$ |
| Data-Matched ERM Control | $86.8_{\pm 5.6}$ | $70.4_{\pm 12.3}$ |
| Linear Probe | $91.7_{\pm 0.6}$ | $74.2_{\pm 3.1}$ |
| ERM (LP-FT) | $68.6_{\pm 4.3}$ | $23.2_{\pm 8.6}$ |
| PRuSC | $62.1_{\pm 3.7}$ | $44.5_{\pm 7.2}$ |
| RoboShot | $73.2_{\pm 0.0}$ | $59.6_{\pm 0.0}$ |
| ZeroShot | $78.5_{\pm 0.0}$ | $56.3_{\pm 0.0}$ |
| *Pair 2: Ptarmigan vs. Prairie-Chicken* | | |
| BAP (Ours) | $\mathbf{81.4}_{\pm 2.2}$ | $\mathbf{65.0}_{\pm 3.7}$ |
| Data-Matched ERM Control | $67.0_{\pm 1.4}$ | $43.9_{\pm 2.4}$ |
| Linear Probe | $80.5_{\pm 1.4}$ | $50.4_{\pm 8.2}$ |
| ERM (LP-FT) | $65.3_{\pm 6.4}$ | $19.2_{\pm 4.9}$ |
| PRuSC | $57.1_{\pm 4.7}$ | $33.7_{\pm 5.2}$ |
| RoboShot | $74.7_{\pm 0.0}$ | $49.7_{\pm 0.0}$ |
| ZeroShot | $81.6_{\pm 0.0}$ | $46.4_{\pm 0.0}$ |

The results in Table 3 demonstrate that the robustness gains facilitated by BAP successfully transfer to natural images. In both evaluations, BAP-pre-trained models yield an improvement of over 10 percentage points in WGA compared to standard LAION-pre-trained encoders. Furthermore, we observe a significant performance gap between BAP and the Data-Matched ERM Control suggesting our anchor extraction and alignment protocol facilitates a degree of background invariance that cannot be attributed to data exposure alone. This consistent performance across both synthetic Waterbirds and real CounterAnimal benchmarks highlights

BAP's utility as a robust feature extractor for practical deployment.

### 6.3. Beyond Birds: NICO++ Vehicle Classification

Following the protocol in Section 6.2, we evaluate BAP within a sim-to-real framework, selecting a novel domain to demonstrate the versatility of our method. This evaluation comprises a series of downstream binary vehicle classification tasks. Specifically, BAP is applied once to pre-train a shared frozen backbone, which then serves as a common feature extractor for independent downstream linear probes. Unlike Sections 6.1 and 6.2, this section uses a ConvNext backbone to demonstrate cross-architectural utility.

To construct the synthetic pre-training data for BAP and the Data-Matched ERM Control, we utilize segmented vehicle instances from the COCO dataset (Lin et al., 2014) composited onto Places365 backgrounds. The following COCO object classes were selected: 'car', 'bus', 'boat', 'train', 'bicycle', and 'airplane'. Notably, we exclude 'truck' instances from the pre-training set to evaluate whether BAP modifies the inductive bias at the super-class level (vehicles).

For the downstream tasks, we employ instances from the NICO++ dataset (Zhang et al., 2023), which presents a significant challenge due to the prevalence of truncated, occluded, or incomplete object instances. Similar to dataset construction in Section 6.2, we construct the training set to have a 100% spurious correlation rate (i.e. all cars appear on grass while all trucks appear on water). The test set is identical in construction to the test set used in Section 6.2.

We show our methods' results along with the most competitive baselines in Table 4 where it is shown that BAP maintains a dominant lead in the first two vehicle pairs. While the performance gap narrows for the Ship vs. Sailboat task, this is primarily attributed to the high baseline robustness of the native linear probe in that specific domain.

We draw attention to the Car vs. Truck task, where BAP yields a **30-percentage-point increase in WGA** over the native CLIP representations. Since 'truck' instances were explicitly omitted from the synthetic pre-training object pool, this result strongly indicates that BAP successfully recalibrates the model's inductive bias toward super-class-wide background invariance rather than category-specific memorization.

### 7. Discussion and Limitations

In this work, we investigate the tendency of CLIP image encoders toward higher levels of spurious background errors relative to their supervised counterparts. We reveal that CLIP represents scenes as near-linear combinations of separate foreground and background elements Section 4 and show that text encoders behave more like "bag-of-features"

*Table 4.* Average and worst-group accuracies (mean $\pm$ std over 5 runs) across the best performing baselines for three pairs of binary vehicle classification on the NICO++ dataset using ConvNeXt-W.

| Method | AVG ($\uparrow$) | WGA ($\uparrow$) |
|---|---|---|
| *Pair 1: Car vs. Truck* | | |
| BAP (Ours) | **83.8** $_{\pm1.4}$ | **72.9** $_{\pm2.0}$ |
| Data-Matched ERM Control | 81.8 $_{\pm0.6}$ | 49.3 $_{\pm2.8}$ |
| Linear Probe | 79.6 $_{\pm2.0}$ | 40.0 $_{\pm7.0}$ |
| ZeroShot | 79.0 $_{\pm0.0}$ | 56.5 $_{\pm0.0}$ |
| *Pair 2: Car vs. Bus* | | |
| BAP (Ours) | **87.0** $_{\pm1.3}$ | **81.5** $_{\pm0.9}$ |
| Data-Matched ERM Control | 84.5 $_{\pm0.3}$ | 63.5 $_{\pm0.4}$ |
| Linear Probe | 81.1 $_{\pm0.9}$ | 43.9 $_{\pm0.5}$ |
| ZeroShot | 82.8 $_{\pm0.0}$ | 58.2 $_{\pm0.0}$ |
| *Pair 3: Ship vs. Sailboat* | | |
| BAP (Ours) | **84.6** $_{\pm0.6}$ | **79.1** $_{\pm1.7}$ |
| Data-Matched ERM Control | 82.4 $_{\pm0.8}$ | 74.8 $_{\pm1.3}$ |
| Linear Probe | 79.9 $_{\pm0.3}$ | 69.3 $_{\pm1.4}$ |
| ZeroShot | 79.4 $_{\pm0.0}$ | 53.7 $_{\pm0.0}$ |

models (e.g., Word2Vec (Mikolov et al., 2013)) than contextual transformers (e.g., BERT (Devlin et al., 2019; Appendix A). Leveraging the fact that foreground representations remain mostly linearly separable from backgrounds, we propose **BAP**, a method of inducing background invariance in CLIP image encoders.

Evaluations across several benchmarks demonstrate that BAP yields superior background invariance and robust OOD generalization compared to existing techniques (Sections 6.1, 6.2, 6.3). Notably, performance does not degrade in the challenging 100% spurious correlation setting where no minority group data is available in training or validation sets. A key practical advantage of BAP is its position as a one-time robustification pre-training step, yielding an encoder that can be deployed to multiple downstream tasks via traditional ERM.

We acknowledge that BAP is more limited in scope than general strategies like DFR or PruSC, as it is specifically formulated for spatially separable correlations and would not function where the spurious feature is part of the core object, such as in CelebA (Liu et al., 2015). However, given the robustness profile exhibited by BAP and the ease of downstream deployment, we believe our method is superior for the specific case of background spurious correlations. Furthermore, we recognize that BAP relies on segmented foreground items which may be difficult to obtain. However, our investigation indicates that a small subset of segmented items is sufficient given certain considerations are made (Appendix F), and we demonstrate that the quality of segmentations need not be perfect for high performance (Appendix D).

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

# A. Appendix: CLIP Text Encoder Additivity

## A.1. Probing Compositional Linear Superposition

To assess whether CLIP's text encoder behaves approximately as a bag-of-words model, we examine whether phrase embeddings are represented as a linear superposition of their constituent word embeddings. Given two content words $w_a$ (foreground) and $w_b$ (background) and an encoder $f(\cdot)$, we embed single-word prompts $s_a, s_b$ and the joint prompt $s_{a,b}$ into unit vectors $e_a, e_b, e_{a,b} \in \mathbb{R}^d$. The degree of additivity is quantified by the cosine similarity:

$$\mathrm{Sim}_{\mathrm{Both,Sum}}(w_a, w_b) = \cos(e_{a,b},\ e_a + e_b).$$

High values of $\mathrm{Sim}_{\mathrm{Both,Sum}}$ indicate that the phrase embedding is well-approximated by the vector sum of its parts, a hallmark of bag-of-words models like Word2Vec.

## A.2. Experimental Setup and Prompt Construction

The evaluation spans 175 foreground–background word pairs (e.g., "bird"–"swamp") chosen to reflect couplings that frequently appear as spurious correlations in vision benchmarks. For CLIP-style encoders, prompts are instantiated using a standard image-centric template:

- $s_a =$ "a photo of a $w_a$"

- $s_b =$ "a photo of a $w_b$"

- $s_{a,b} =$ "a photo of a $w_a, w_b$"

For non-CLIP baselines (Word2Vec and BERT), raw word strings are used (e.g., $s_a = w_a$) to avoid injecting template artifacts foreign to their pretraining domains. All resulting embeddings are $\ell_2$-normalized such that $\|e_a\|_2 = \|e_b\|_2 = \|e_{a,b}\|_2 = 1$. We report both the compositional additivity ($\mathrm{Sim}_{\mathrm{Both,Sum}}$) and the lexical similarity between the words themselves ($\mathrm{sim}_{\mathrm{T1,T2}}(w_a, w_b) = \cos(e_a, e_b)$).

*Table 5.* Mean cosine similarities ($\pm$ standard deviation) across 175 word pairs. CLIP encoders exhibit high additivity scores (0.90–0.91), nearly matching the Word2Vec bag-of-words baseline and significantly exceeding the contextual BERT baseline.

| | WORD2VEC (BASELINE) | | CLIP-VIT-B/16 | | CLIP-CONVNEXT | | BERT (CONTEXTUAL) | |
|---|---|---|---|---|---|---|---|---|
| **METRIC** | $\mathrm{Sim}_{\mathrm{T1,T2}}$ | **$\mathrm{Sim}_{\mathbf{B,S}}$** | $\mathrm{Sim}_{\mathrm{T1,T2}}$ | **$\mathrm{Sim}_{\mathbf{B,S}}$** | $\mathrm{Sim}_{\mathrm{T1,T2}}$ | **$\mathrm{Sim}_{\mathbf{B,S}}$** | $\mathrm{Sim}_{\mathrm{T1,T2}}$ | **$\mathrm{Sim}_{\mathbf{B,S}}$** |
| SCORE | $0.03 \pm 0.08$ | $\mathbf{0.99 \pm 0.01}$ | $0.61 \pm 0.06$ | $\mathbf{0.90 \pm 0.03}$ | $0.70 \pm 0.06$ | $\mathbf{0.91 \pm 0.02}$ | $0.66 \pm 0.12$ | $0.82 \pm 0.03$ |

## A.3. Results and Relation to Spurious Bias

As a baseline, Word2Vec is almost perfectly additive ($\mathrm{Sim}_{\mathrm{Both,Sum}} \approx 1.0$), while BERT is substantially less so ($0.82 \pm 0.03$). This discrepancy reflects BERT's deep contextualization and self-attention mechanism, where the embedding of a phrase like "duck, pond" is a non-linear, contextually enriched representation rather than a simple superposition of constituents.

In contrast, CLIP's text encoders remain highly additive ($0.90 \pm 0.03$ for ViT-B/16 and $0.91 \pm 0.02$ for ConvNeXt), nearly matching the bag-of-words behavior of Word2Vec. This suggests that CLIP's contrastive training objective encourages the detection of salient keywords over the modeling of complex phrasal structure.

This near-linear superposition has direct implications for background bias. When $e_{a,b}$ is well-approximated by $e_a + e_b$, foreground and background components occupy nearly independent axes in the text embedding space. Consequently, the background direction remains prominent and can be reused across disparate objects (e.g., "bird in swamp" vs. "dog in swamp"). In the multimodal alignment space, images sharing a background but differing in object will likely end up close to one another whenever the background component is strong, facilitating CLIP's reliance on backgrounds as classification shortcuts. These results mirror our findings on the image-encoder side, Section 4, suggesting a consistent mechanistic behavior where CLIP represents scenes as a "bag of features" across both modalities.

# B. Embedding Space Analysis

## B.1. Background Sensitivity Index (BSI)

In order to assess the effect of our method on the CLIPs embedding space we introduce a proprietary metric aimed at quantifying the degree of background invariance.

The Background Sensitivity Index (BSI) quantifies an image encoder's sensitivity to background changes by measuring the shift in object representations relative to their natural embedding variance. For a fixed object class, we partition the image encoders' embeddings $\mathbf{z}$ by background into sets $A$ and $B$, compute centroids $\boldsymbol{\mu}_A, \boldsymbol{\mu}_B$ and variances $\sigma_A^2, \sigma_B^2$, then calculate

$$\text{BSI} = \frac{\|\boldsymbol{\mu}_A - \boldsymbol{\mu}_B\|_2}{\sqrt{\sigma_A^2 + \sigma_B^2}}.$$

Specifically, for the Waterbirds benchmark, we calculate this index by isolating the impact of the background on each species' representation. For the landbird class, we define set $A$ using using landbird images placed against land backgrounds and set $B$ using the same bird images with the land backgrounds swapped for water ones. We then compute the respective centroids and variances ($\boldsymbol{\mu}_A, \sigma_A^2$ and $\boldsymbol{\mu}_B, \sigma_B^2$) for these sets and apply the BSI equation. This process is repeated for the waterbird class, and the results are averaged to provide a unified measure of how much the background swap, regardless of the bird itself, shifts the model's internal representations.

*Table 6.* Background Sensitivity Index (BSI) (mean $\pm$ std) across a variety of baselines. The results in this table are an extension of the results presented in Table 2 and are therefore grouped similarly.

| Method | Waterbirds-95% (Original) BSI($\downarrow$) | Waterbirds-100% BSI ($\downarrow$) |
|---|---|---|
| ZeroShot | $30.4_{\pm 0.0}$ | $30.4_{\pm 0.0}$ |
| RoboShot | $30.4_{\pm 0.0}$ | $30.4_{\pm 0.0}$ |
| Linear Probe | $30.4_{\pm 0.0}$ | $30.4_{\pm 0.9}$ |
| AFR | $37.5_{\pm 1.0}$ | - |
| DFR | $37.8_{\pm 0.9}$ | - |
| ERM (LP-FT) | $38.5_{\pm 1.2}$ | $82.9_{\pm 2.1}$ |
| ERM + RandAugment | $37.9_{\pm 1.0}$ | $75.9_{\pm 2.7}$ |
| WiSE-FT | $32.8_{\pm 0.1}$ | $47.3_{\pm 0.5}$ |
| PruSC | $31.4_{\pm 2.1}$ | $34.9_{\pm 1.7}$ |
| Data-Matched ERM Control | $9.8_{\pm 0.8}$ | $10.3_{\pm 0.7}$ |
| **BAP** (Ours) | $\mathbf{2.2}_{\pm 0.1}$ | $\mathbf{2.4}_{\pm 0.3}$ |

Low BSI values indicate strong background invariance, where centroid shifts remain small compared to intra-group spread; high values reveal notable sensitivity, where background changes drive substantial embedding shifts. As can be seen in Table 6, BAP facilitates a level of background invariance that is unmatched by any of the other baselines. Our approach is an order of magnitude lower in background sensitivity compared to CLIP's native representations and nearly three times lower than the Data-Matched ERM control, further indicating that our anchor generation and alignment strategy facilitates background invariance in a manner not possible using standard cross-entropy.

To visualize the effect that BAP has on the embedding space, we plot UMAP visualizations for isolated foregrounds, backgrounds, and composite scenes for both native CLIP embeddings and post-BAP embeddings in Figure 4. Most notably, in the full scene embeddings, the original CLIP representation shows a hierarchical grouping where images are first grouped by background and then sub-grouped by bird class. However, post-BAP embeddings indicate that the separation is made almost entirely based on the foreground class.

Note that the separation in BAPs background-only embeddings is significantly decreased from the original CLIP embeddings. Our investigation reveals that BAP negatively impacts background and scene recognition, for further details, see Appendix G.

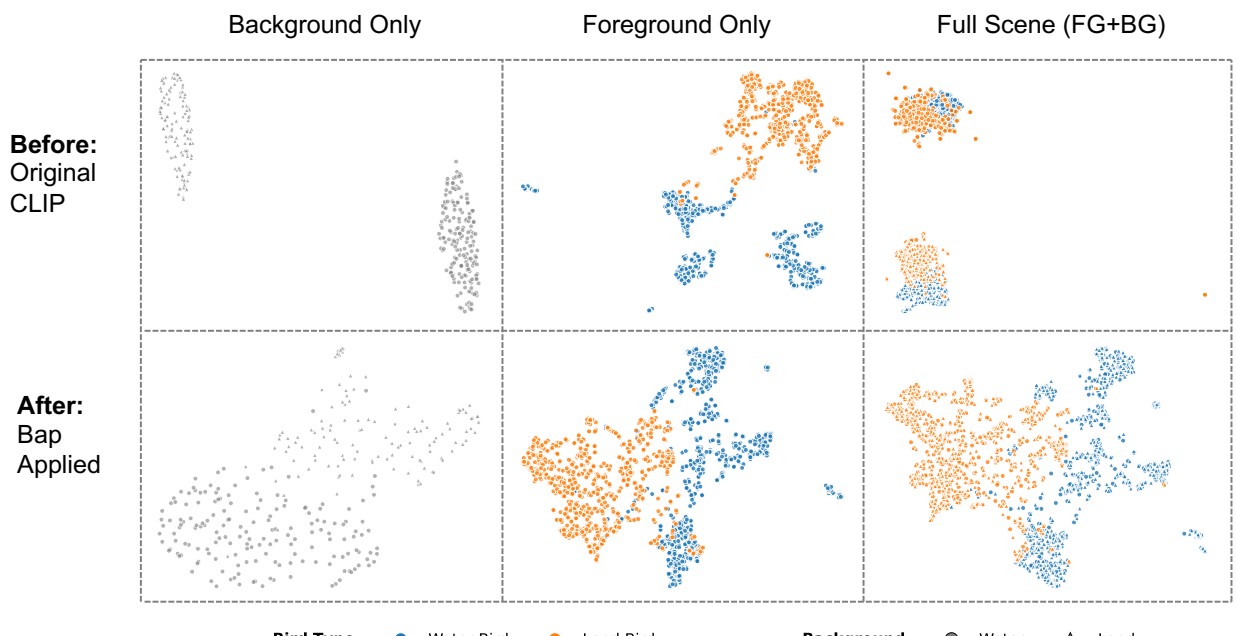

*Figure 4.* **UMAP visualizations of an image encoders embedding space before and after BAP.** The top row contains UMAP visualizations for CLIP's embedding space while the bottom row contains visualizations after BAP application. From left to right we have: background-only embeddings, foreground-only embeddings, and full scene ( foregrounds composited onto backgrounds ) embeddings.

## C. Appendix: Ablation on Anchor Semantic Necessity

### C.1. Random Orthogonal Target Assignment

In the standard BAP formulation, Phase 1 is designed to extract a set of foreground-only anchor vectors using a frozen teacher encoder. This process aims to preserve the rich semantic structure of foreground objects while suppressing background noise. To isolate whether the observed robustness gains are driven by these high-fidelity semantic anchors or simply by the geometric enforcement of invariance, we conducted an ablation study using random orthogonal training targets.

This ablation entirely removes **Phase 1: Anchor Vector Extraction**. Instead of computing unique, instance-specific anchor vectors (**a**) by averaging composites, we initialize two static, synthetic target vectors: $\mathbf{v}_{water}$ and $\mathbf{v}_{land}$.

- Initialization: The vectors $\mathbf{v}_{water}$ and $\mathbf{v}_{land}$ are generated as random orthogonal vectors in the embedding space $\mathbb{R}^d$ and normalized to the unit hypersphere such that $\|\mathbf{v}\|_2 = 1$ and $\mathbf{v}_{water} \cdot \mathbf{v}_{land} = 0$.

- Target Assignment: We collapse the granular species-level distinctiveness of the training targets. All foreground instances belonging to the Waterbird super-class are assigned $\mathbf{v}_{water}$ as their target, while all Landbird instances are assigned $\mathbf{v}_{land}$.

- Alignment Training: We then proceed to **Phase 2: Robust Alignment Pre-training**. The student encoder $f_\theta$ is optimized to minimize the cosine distance between the embedding of a randomized synthetic composite and its assigned super-class vector:

$$\mathcal{L}_{align} = 1 - \cos(f_\theta(\hat{x}), \mathbf{v}_{target}) \qquad (4)$$

This configuration maintains the same "virtual epoch" size and data exposure as the standard BAP protocol. However, it explicitly discards the learned semantic foreground signal of specific species (e.g., "Black-footed Albatross") in favor of a coarse super-class grouping. By comparing this baseline against standard BAP, we can quantify the degree to which robustness relies on the high-fidelity distillation of the foreground signal versus the simple, forced suppression of background

features. For this ablation, we used the exact same CUB-Places365 paradigm as in Sections 6.1 and 6.2 to conduct pre-training. Following this, we train various linear probes on a number of downstream data sets, again identical to the two-stage we employ all throughout; we tested downstream performance on the Waterbirds benchmark followed by the two CounterAnimal pairs we used in Section 6.2. Note that we consider the Waterbirds task to be an in-distribution (I.D) task, since the same CUB-Places365 paradigm is used in both training and testing, while we consider the CounterAnimal tasks as OOD tasks.

*Table 7.* Performance Metrics for Random Orthogonal Target Alignment. We report Average Accuracy and Worst Group Accuracy across different evaluation tasks (mean $\pm$ std over 3 runs).

| Task | Average Accuracy | Worst Group Accuracy |
|---|---|---|
| **Waterbirds** | $93.5_{\pm 0.5}$ | $90.2_{\pm 0.4}$ |
| **CounterAnimal**: Bulbul vs. Brambling | $23.1_{\pm 4.1}$ | $11.7_{\pm 3.7}$ |
| **CounterAnimal**: Ptarmigan vs. Prairie-Chicken | $14.7_{\pm 5.3}$ | $7.1_{\pm 2.7}$ |

The results in Table 7 tell a very compelling story: for the I.D task (Waterbirds) we see nearly identical performance to the full BAP implentation in Table 2, however, performance collapses catastrophically for the OOD CounterAnimal tasks.

The success of this ablation on the I.D Waterbirds task indicates that the mere projection of all Waterbird and Landbird instances to single fixed vectors is sufficient to suppress the various background signals. Indeed, we believe that this behavior can best be understood through the lens of an information bottleneck and by analogy to a $\beta$-VAE (Higgins et al., 2017). In a $\beta$-VAE, a weighted KL-divergence term creates a tight information bottleneck that forces the model to encode only the factors of variation necessary to reconstruct the shared structure of the data, effectively pruning out non-essential details. Similarly, the $\mathcal{L}_{align}$ objective enforces a strict many-to-one mapping. By constraining the model to map several distinct background variations of the same object to a singular, invariant point a in the embedding space, we artificially induce a bottleneck. To minimize the loss, the encoder is forced to discard the high-variance, spurious background information and encode only the salient, common factors pertaining to the birds in this instance.

The catastrophic performance, however, in the OOD tasks, makes this ablated version of BAP unusable in the real world. We believe that the drop in performance is due to two factors: firstly, the above formulation's information bottleneck is restrictive to the point of potentially causing catastrophic forgetting where the image encoder is incapable of recognizing OOD samples. Secondly, our formulation of the anchor vectors preserves Clip's rich semantic instance-specific information, while this version discards it leading to a potentially irregular embedding space that is void of the rich semantic structure that CLIP is known for.

The above ablation isolates the necessity of different aspects of BAP:

1. Mapping several instances of a given foreground, on randomized backgrounds, to a single point in the embedding space is responsible for the suppression of background signals and produces the observed quality of background invariance.

2. Utilizing the frozen CLIP image encoder as a teacher model to create individualized anchor vectors preserves CLIP's rich semantic information which allows the increased robustness to transfer to OOD domains. Furthermore, this aspect facilitates a more gentle restructuring of the embedding space, and by extension the internal model representations, such that catastrophic forgetting is mitigated.

## D. Ablation Study: Sensitivity to Segmentation Quality

To assess the robustness of our method against imperfect data preprocessing, we conducted a sensitivity analysis on the quality of the segmentation masks used during the synthetic composite generation. While our primary experiments utilize the high-quality, human-annotated segmentation masks provided by the CUB dataset, real-world deployment scenarios often rely on automated segmentation models which may produce noisy or coarse outputs. We therefore replicated BAP across four distinct levels of segmentation fidelity to measure the downstream impact on Worst Group Accuracy (WGA):

- **Perfect (Baseline):** We utilize the standard, fine-grained segmentation masks provided by the CUB dataset. This represents the ideal scenario where the object is perfectly isolated from its original background.

- **Noisy (Dilated):** To simulate automated segmentation masks that fail to tightly contour the object, we apply a morphological dilation (radius = 15px) to the original mask. This results in a "halo" effect, where the alignment model is exposed to the bird along with a significant margin of pixels from the original, potentially spurious, background.

- **Botched (Eroded):** To simulate over-aggressive filtering or partial occlusion, we apply a morphological erosion (radius = 21px) to the mask. This removes peripheral features of the bird (e.g., beaks, tails, and crests), forcing the model to align based on incomplete semantic information.

- **Bounding Box:** We replace the fine-grained mask with a rectangular bounding box derived from the mask's extents. This represents the coarsest possible segmentation, where the object is pasted along with all local background context contained within the box, serving as a lower-bound baseline for segmentation precision.

By comparing performance across these variations, we aim to determine if our method requires pixel-perfect isolation of the target object or if it remains effective even when the synthesis process includes spurious background noise or incomplete object features.

*Table 8.* Sensitivity Analysis of Segmentation Quality. We report Average Accuracy and Worst Group Accuracy (WGA) across different segmentation degradation modes (mean $\pm$ std over 3 runs).

| Segmentation Mode | Average Accuracy | Worst Group Accuracy |
|---|---|---|
| **Perfect** | $94.12_{\pm 0.6}$ | $91.80_{\pm 0.5}$ |
| **Noisy** | $93.63_{\pm 0.5}$ | $89.98_{\pm 1.7}$ |
| **Botched** | $92.66_{\pm 0.6}$ | $90.52_{\pm 0.4}$ |
| **Bounding Box** | $89.83_{\pm 0.7}$ | $82.53_{\pm 1.3}$ |

The results of this ablation study, detailed in Table 8, reveal a clear hierarchy of performance correlated with segmentation fidelity, yet demonstrate a surprising degree of robustness. As expected, the **Perfect** segmentation yields the highest Worst Group Accuracy (91.80%), serving as our upper bound. Notably, the **Botched** (eroded) variation outperforms the **Noisy** (dilated) variation in WGA (90.52% vs. 89.98%), with significantly lower variance. This suggests that for robust alignment, it is preferable to sacrifice peripheral object details (via erosion) rather than risk including spurious background features (via dilation). Finally, while the **Bounding Box** approach suffers a significant drop in performance (82.53% WGA), it remains functional, indicating that the method can still extract useful semantic alignment signals even from coarse, unsegmented localization data, though removal of the original background remains critical for optimal performance.

# E. Foreground-only Anchor Generation

To validate our anchor generation procedure, we utilize the text encoder as a semantic ground truth. Our objective is to determine if the proposed 'purification' process—averaging embeddings of a fixed foreground across varying backgrounds—successfully suppresses background noise while enhancing foreground signal. The experimental protocol iterates through a diverse set of bird species, selecting multiple unique image instances per species to ensure robustness. For each image instance, we generate a sequence of anchor vectors by compositing the segmented foreground onto $N$ randomly sampled background images. We then compute the mean of the resulting image embeddings to form a single 'purified' vector for that specific value of $N$. To quantify the quality of this vector, we calculate its cosine similarity to a text embedding generated from the prompt template 'A photo of a {}' where we insert the species name into the braces.

We hypothesize that as $N$ increases, the incoherent background signals will cancel out via averaging, leaving a stable foreground representation. Consequently, we expect to observe a monotonic increase in cosine similarity between the image anchor and the text prompt, indicating that the anchor is becoming a cleaner representation of the species.

As can be seen in Figure 5, our hypothesis holds true as the cosine similarity displays a rapid climb as the number of novel background Contexts are used in constructing the anchor vector. While we do see an increase in relative cosine similarity score, the absolute values might seem concerning at first glance, given that cosine similarity occupies a range of $[-1, 1]$ with values nearer to 1 indicating higher agreement. A naive reading of Figure 5's peak cosine similarity file ($< 0.5$) might lead us to believe that there is overall low agreement between the image and corresponding text embeddings.

However, recent work (Papadimitriou et al., 2025) indicates that CLIP image and text embeddings occupy separate regions in the shared embedding space, and that relative alignment between images and corresponding text prompts is more significant

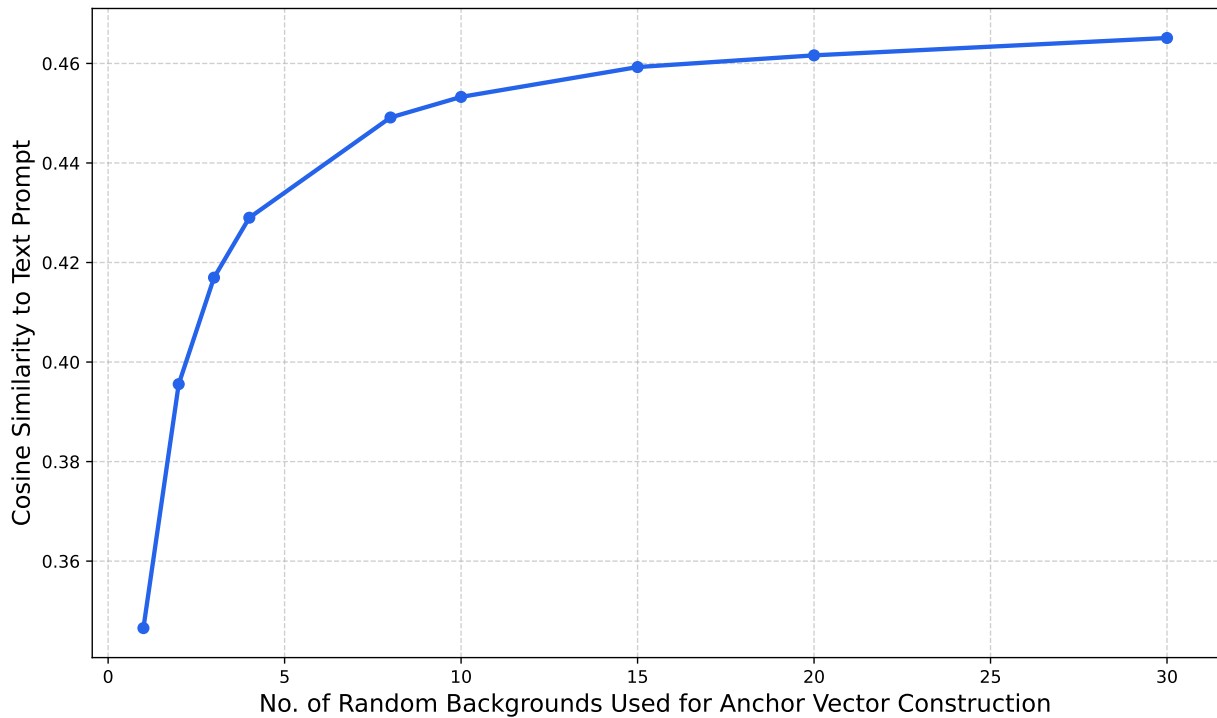

*Figure 5.* **Cosine similarity between text prompt embedding and anchor vector.** The x-axis indicates how many novel foreground-background pairings were selected for anchor vector construction. The y-axis indicates the average cosine similarity between the anchor vectors and their corresponding species-specific text prompts.

than absolute alignment. Therefore, the increase from around $0.35$ to $0.46$ in Figure 5, representing a $30\%$ increase in absolute similarity score, is clear proof that our method of anchor generation amplifies the foreground signal by canceling out the various random background signals.

## F. Data Requirements

This section aims to assess how data-hungry BAP is and what the overall data requirements are in order to obtain optimal performance.

### F.0.1. HOW MANY SEGMENTED FOREGROUNDS ARE NEEDED?

Firstly, we study the relationship between the number of distinct foreground items, used during the second phase of BAP (alignment), and performance. Typically, each foreground item would be repeatedly composited onto a number of randomly selected backgrounds. However, in order to independently assess the effect of the total number of foreground items used, we opt to only repeat each foreground item once for the purposes of this investigation.

Figure 6, shows a rapid initial increase in worst-group accuracy, followed by a much more gradual increase to peak performance. Quite surprisingly, we find that only 500 total segmented birds was sufficient to reach $90\%$ of peak worst-group accuracy. This leads us to believe that in addition to BAP's highly practical deployment potential, it is not particularly data-hungry and would perform reasonably well with a modest number of segmented foreground items.

Furthermore, we note the performance levels at around $3,000$ total foreground items nearly matches the performance reached in Table 2 where the total number of birds available in the Waterbirds training set were utilized. Additionally, each bird repeatedly composited onto four distinct random backgrounds; this setup is clearly overkill given the performance we attained with the truncated and non-repeated dataset.

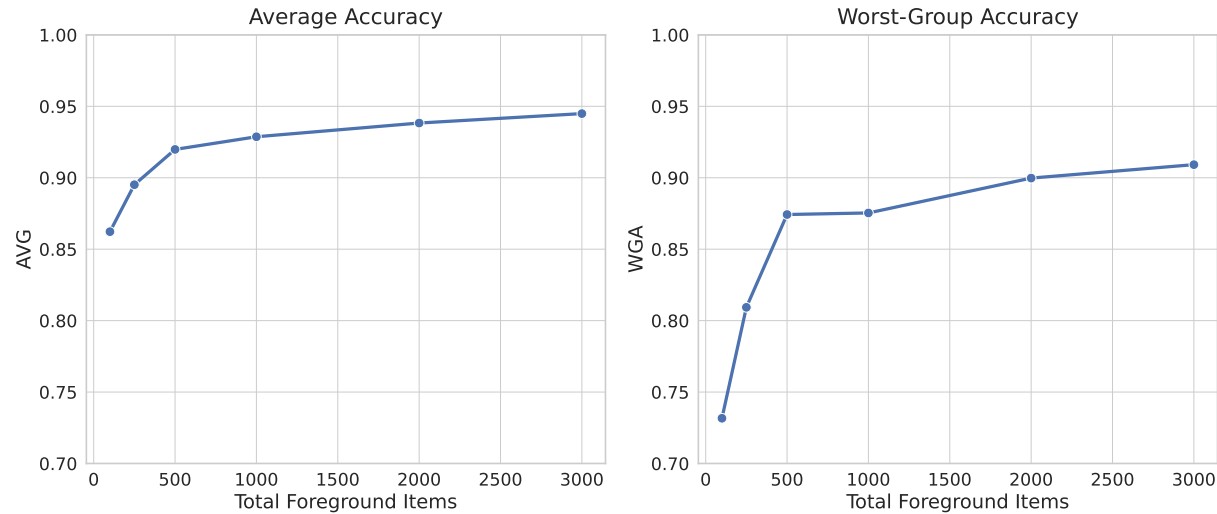

*Figure 6.* **Performance vs Total Segmented Foregrounds Used.** Above, we investigate the effect of the number of selected foregrounds (birds) used by BAP on performance on the Waterbirds benchmark. The x-axis indicates how many total birds were sampled for each BAP run and the y-axes indicates average and worst group accuracy performance. We observe WGA increases sharply when going from 100 to 500 total birds but then plateaus and increases slowly before reaching peak performance at around 3,000 total birds.

### F.0.2. BACKGROUND RANDOMIZATION REQUIREMENTS — PHASE 1(ANCHOR VECTOR GENERATION)

To assess the effect of the number of background randomizations used in the anchor generation phase directly on model performance (rather than via similarity with corresponding text embedding as in Appendix E), we construct a simple hyperparameter sweep. We vary the number of novel background contexts: each foreground item appears against during the anchor vector construction phase and proceeds with BAP using CUB-Places 365 composites and the Waterbirds benchmark for linear probe training and testing. Essentially, we are varying the $K$ parameter in Equation (5.1.1) to assess its effect on training performance.

*Table 9.* **Sensitivity Analysis of K.** We report Average Accuracy and Worst Group Accuracy across different values of K to determine the number of background contexts required in anchor vector generation for optimal performance .

| $K$: No. of Backgrounds | Average Accuracy | Worst Group Accuracy |
|:---:|:---:|:---:|
| 1 | $93.5_{\pm 0.8}$ | $87.1_{\pm 1.2}$ |
| 2 | $93.9_{\pm 0.7}$ | $89.2_{\pm 0.8}$ |
| 4 | $94.1_{\pm 0.6}$ | $90.9_{\pm 0.7}$ |
| 8 | $94.7_{\pm 0.5}$ | $91.8_{\pm 0.8}$ |
| 16 | $94.5_{\pm 0.5}$ | $92.1_{\pm 0.9}$ |

We observe in Table 9 a predictable pattern of increased random foreground-background pairings (K) resulting in higher robustness and more stable performance. These results are in line with observations from Appendix E where we demonstrated that higher k values resulted in higher cosine similarity scores with the corresponding species-level text prompt. These findings indicate that as k is increased, the foreground signal is amplified while the background signals are attenuated.

### F.0.3. BACKGROUND RANDOMIZATION REQUIREMENTS — PHASE 2 (ALIGNMENT PRE-TRAINING)

The results of the preceding section perhaps render the repetition of each foreground item (in phase 2) against a number of randomly selected backgrounds somewhat moot given that performance reached near peak levels with each foreground item appearing only against one background item. Indeed, studying the effect of increased background randomization at higher total foreground numbers would not yield productive insights since performance is already saturated. Therefore, we investigate the effect of the number of background randomizations at lower total foreground counts (100 and 250 total

foreground items selected). This is an excellent opportunity as well to investigate how performance can be maximized in the absence of a large number of segmented foreground objects

To clarify, the total number of foregrounds indicates how many distinct birds were used while the background randomizations indicate how many times each foreground was composed onto a randomly selected background. For instance, if we select 1000 foreground items along with 4 background randomizations for each item, then our effective dataset size becomes 4000 total images.

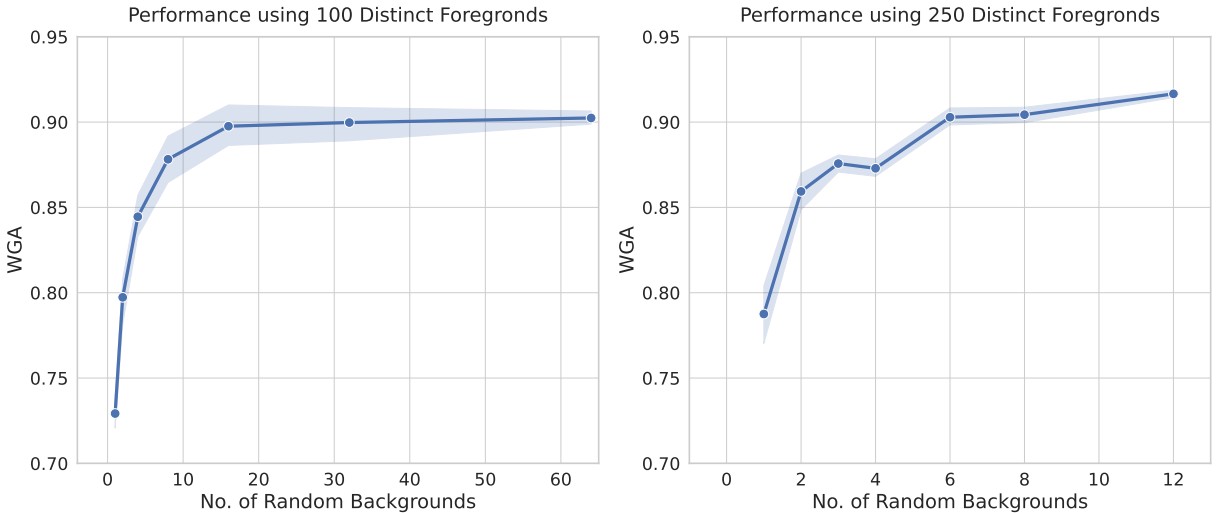

*Figure 7.* **Performance vs Number of Background Randomizations.** The above plot outlines the effect of the number of random backgrounds each foreground item is placed on during Phase 2 of BAP on worst-group accuracy performance. We use 100 and 250 distinct foreground items (left then right).

In Figure 7, we observe a predictable but interesting trend: firstly, as the number of distinct foregrounds increases, the number of background randomizations needed to reach peak WGA performance decreases. Furthermore, and most surprisingly, we are able to attain near peak performance with only 100 distinct foreground items by radically increasing the number of background randomizations ($\geq 16$). This further increases the practicality of BAP: in settings where access to high-quality segmented foreground items is limited, we may resort to radically increasing the number of background randomizations to attain higher levels of performance.

The above performance indicates that BAP is a flexible algorithm as regards data requirements which can be fluid without compromising near peak performance. Depending on the availability of segmented foreground items, users may seek to use a low number of background randomizations if an abundance of segmented foreground items is available, or increase the number of background randomizations otherwise.

# G. BAPs Impact on Scene Recognition and Limitations Thereof

To understand how BAP affects the model's broader capabilities, we designed a zero-shot evaluation framework. This allowed us to compare the original, pre-trained CLIP vision encoder directly against our aligned version without the need for additional fine-tuning or linear probes. Our goal was to assess performance across two distinct areas: domain-specific object classification (Waterbirds test set) and open-domain scene classification (Places365 backgrounds).

## G.1. Zero-Shot Classifier Construction

For all zero-shot tasks, we froze the text encoder and generated classifier weights using a standard prompt ensembling technique. Rather than relying on a single text prompt, we created a robust "prototype" for each class by inserting the class name into 40 distinct templates (e.g., "a photo of a { }", "a bad photo of a { }").

We encoded these 40 variations and averaged their embeddings to create a single representative vector for each class. During inference, classification was performed by comparing the image embedding to these class prototypes; the model simply

predicted the class whose prototype had the highest similarity score with the image. We performed this procedure twice: once using the original CLIP vision weights and once using our aligned weights.

### G.2. Evaluation Tasks

We evaluated the models on two distinct datasets to measure both the improvement in object focus and the retention of background knowledge.

#### G.2.1. TASK A: ROBUST OBJECT CLASSIFICATION (WATERBIRDS)

The primary goal of this task was to assess whether the alignment allows the model to better separate the in-class images in a zero-shot capacity. Given our results in Section 6.1, we expect performance to be higher on the BAP model than the native CLIP model.

After applying BAP using the same CUB-Places365 composites used in Section 6.1, we evaluated the model on the official Waterbirds test set using a simple binary classification scheme ("landbird" vs. "waterbird").

#### G.2.2. TASK B: BACKGROUND RETENTION (PLACES)

This task investigates BAPs effect on CLIPs general-purpose, zero-shot classification abilities. We have already demonstrated that, in the presence of the core foreground signal, BAP image encoders ignore most background variation. However, what we seek to understand with this test is BAP's effect on the image encoders representations of backgrounds and out-of-class objects *in the absence of core foreground items*.

We aggregated images from the Places365 dataset corresponding to the background categories used during training (e.g., *ocean, attic, igloo*). We then performed multi-class classification to verify if the model could still correctly identify these scenes when explicitly queried. The Places365 Dataset offers a comprehensive supply of both general background scenes (e.g., ocean, rainforest) and more object centered scenes (e.g., igloo, windmill).

The results in Table 10 are dramatic: we observe higher zero-shot classification scores for our target class (birds) using BAP, however, we note a catastrophic collapse in the BAP encoders ability to classify out-of-class scenes and objects. This is expected given the embedding space visualizations seen in Figure 4.

Given that BAPs 'raison d'être' is to induce background invariance and isolate model attention to the desired class objects, this behavior is again, expected and further points to our method successfully modifying the encoders inductive bias. By treating the background as a nuisance variable during the anchor generation phase, the model transitions from a context-dependent estimator to an intrinsic object recognizer

This renders the adoption of BAP in real-world scenarios a nuanced task-specific decision; in instances where background invariance is of high practical utility, BAP makes an excellent selection. However, in instances where environmental context acts as a vital signal, BAP may compromise performance. Therefore, BAP is most effectively deployed in specialized scenarios where background invariance is a high-priority requirement for robustness. For instance, in wildlife monitoring and camera trap applications, BAP prevents the model from utilizing specific rocky terrains or forest clearings as "shortcuts" for species identification, ensuring that animals are recognized by their physical characteristics even when they move to novel environments. Similarly, in medical diagnostic imaging, BAP can induce invariance to scanner artifacts or healthy tissue textures, forcing the encoder to focus exclusively on critical clinical markers.

On the other hand, in tasks such as autonomous driving, the background provides the necessary spatial context to disambiguate objects; a red octagon only becomes a "stop sign" when located within a traffic scene. Likewise, in action recognition, the surrounding environment is often essential for full comprehension, as the same physical movement may be interpreted as "playing golf" or "hailing a taxi" based entirely on whether the background is a golf course or a city street. Ultimately, while the "catastrophic collapse" of background classification scores in our results confirms the successful induction of background invariance, it highlights that BAP is best suited for tasks where isolating the target from its surroundings is the primary objective.

*Table 10.* **Zero-Shot Classification Results** We compare the image encoder ability to classify in a zero-shot regime for both native CLIP and post BAP CLIP. We investigate performance both within the target class and on out-of-class scenes and objects in order to assess the impact of BAP on the image encoder's ability to recognize and represent background information.

| Class Name | Original CLIP Accuracy | BAP CLIP Accuracy |
|---|---|---|
| *Task A: Waterbirds (Target Class)* | | |
| Landbird | 0.76 | **0.94** |
| Waterbird | 0.68 | **0.75** |
| *Task B: Places365 (Out-of-class scenes and objects)* | | |
| Abbey | **0.79** | 0.13 |
| Alley | **0.95** | 0.65 |
| Attic | **0.92** | 0.02 |
| Auditorium | **0.89** | 0.04 |
| Bar | **0.92** | 0.16 |
| Bridge | **0.92** | 0.57 |
| Canyon | **0.92** | 0.01 |
| Castle | **0.76** | 0.38 |
| Cemetery | **0.96** | 0.36 |
| Chalet | **0.92** | 0.45 |
| Classroom | **0.93** | 0.00 |
| Closet | **0.98** | 0.05 |
| Crevasse | **0.89** | 0.02 |
| Driveway | **0.83** | 0.11 |
| Engine Room | **0.96** | 0.07 |
| Iceberg | **0.85** | 0.24 |
| Igloo | **0.93** | 0.81 |
| Martial Arts Gym | **1.00** | 0.61 |
| Ocean | **0.81** | 0.01 |
| Pond | **0.90** | 0.41 |
| Rainforest | **0.79** | 0.05 |
| Shopfront | **0.96** | 0.10 |
| Sky | **0.69** | 0.60 |
| Windmill | **0.97** | 0.70 |

# H. Training Progression and Fine-Tuning Constraints

This section gives an account of the progression of the alignment objective $\mathcal{L}_{align}$ during BAP as well as the behavior of the model on the downstream Waterbirds benchmark.

As far as the alignment objective is concerned, initially, we see a sharp decline in loss during the first few epochs, which gradually levels out to approach an asymptote of around 0.05. There is not much insight there beyond the fact that BAP is relatively stable during training. Due to this stable behavior, early stopping on $\mathcal{L}_{align}$ is recommended to reduce unneeded compute resources.

As far as post-BAP dowstream accuracy is concerned, we observe that peak performance is reached after only around 5 epochs of linear probe training. This is not unexpected as BAP produces well-organized, semantically grounded representations.

Following 30 epochs of linear probe training, we unfreeze the weights of the image encoder and fine-tune Using standard cross-entropy on the Waterbirds training set. We note that performance, particularly as regards worst group accuracy, degrades sharply after the weights of the model are fine-tuned on the spuriously correlated Waterbirds training set. This places a critical restriction on downstream deployment since fine-tuning the image encoder in this manner undoes BAP's induced background invariance. Therefore, we strongly recommend that any post-BAP downstream deployment utilize the image encoder as a fixed feature extractor. Given the strong sim-to-real and OOD performance exhibited in Sections 6.2 and 6.3, and given CLIPs native, rich semantic representational ability, we do not anticipate this to be a major limitation of our method.

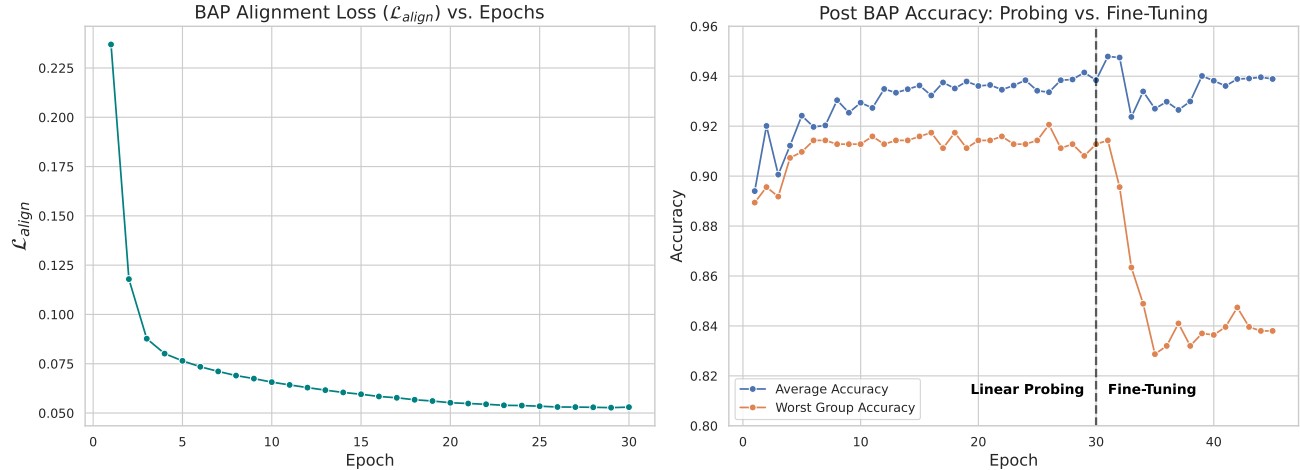

*Figure 8.* Left: progression of the alignment loss during BAP. Right: progression of average and worst group accuracies during linear probing and then full fine-tuning post-BAP.

# I. Appendix: Detailed Implementation Specs

## I.1. Waterbirds Results

To ensure reproducibility, we detail the exact hyperparameters, architectural choices, and training protocols used for the Waterbirds experiments reported in Section 6.1. All experiments were conducted using the PyTorch framework.

### I.1.1. HARDWARE AND ENVIRONMENT

All experiments were executed on a single NVIDIA L40S GPU. We utilized `OpenCLIP` for model instantiation. To ensure deterministic behavior, the random seed was fixed to 42 for Python, NumPy, and PyTorch. We enabled `cudnn.benchmark` to optimize convolution performance and disabled OpenCV threading to prevent deadlock during data loading.

### I.1.2. MODEL ARCHITECTURE AND INITIALIZATION

For all Vision Transformer experiments, we utilized the `ViT-B/16` architecture.

- **Pre-trained Weights:** We initialized the model using the `laion2b_s34b_b88k` weights (trained on the LAION-2B dataset) rather than the original OpenAI weights, as provided by the OpenCLIP library.

- **Input Resolution:** All images were resized to $224 \times 224$ pixels.

- **Normalization:** We applied standard CLIP normalization with mean $(0.4814, 0.4578, 0.4082)$ and standard deviation $(0.2686, 0.2613, 0.2757)$.

### I.1.3. BAP PHASE 1: ANCHOR VECTOR EXTRACTION

In the first phase of Background-invariant Anchor Pre-training (BAP), we extracted robust anchor vectors for the CUB dataset classes.

- **Data Source:** We utilized a subset of 4000 samples from the CUB dataset, ensuring that this subset overlaps with the bird instances in the Waterbirds training set and does not overlap with birds in the validation or test sets.

- **Backgrounds:** We utilized a set of 34 distinct background categories from Places365 (e.g., 'coast', 'swamp', 'forest') for compositing.

- **Composite Generation:** For each bird instance, we generated $N = 10$ randomized composites. The foreground bird was scaled to approximately $0.8\times$ of the target resolution ($224 \times 224$). We applied a threshold mask ($> 100$) and a Gaussian blur ($\sigma = 1$) to the segmentation mask to smooth artifacts.

- **Vector Calculation:** The $N$ composites were encoded using the frozen `ViT-B/16` encoder. The resulting embeddings were mean-pooled and $L_2$ normalized to produce the final instance-specific anchor.

### I.1.4. BAP PHASE 2: ALIGNMENT PRE-TRAINING

We fine-tune the student encoder for **50 epochs** using the following hyperparameters:

- **Batch Size:** 256

- **Learning Rate:** $5 \times 10^{-6}$

- **Optimizer:** AdamW with a weight decay of $0.01$.

- **Dataset Expansion:** We employ a "virtual epoch" strategy where the dataset length is artificially expanded by a factor of $3\times$. This ensures that in every epoch, each unique foreground object is aligned against three distinct, randomly sampled backgrounds, thereby stabilizing the gradients.

- **Scheduler:** We utilize a sequential learning rate scheduler. The learning rate is warmed up linearly from $1\%$ to $100\%$ of the base rate over the first $10\%$ of training steps (Linear Warmup), followed by a Cosine Annealing decay schedule down to a minimum learning rate of $1 \times 10^{-6}$ for the remainder of training.

*Table 11.* Hyperparameters for BAP Phase 2 (Alignment).

| Hyperparameter | Value |
|---|---|
| Loss Function | Cosine Embedding Loss (Margin 1.0) |
| Optimizer | AdamW |
| Batch Size | 256 |
| Learning Rate (Backbone) | $5 \times 10^{-6}$ |
| Weight Decay | 0.01 |
| Epochs | 30 |
| LR Scheduler | Linear Warmup (10%) $\rightarrow$ Cosine Annealing |
| Virtual Epoch Expansion | $3\times$ (Dynamic background randomization) |

## I.2. Downstream Evaluation (Waterbirds)

Following BAP, we evaluated robustness on the real-world Waterbirds dataset using Linear Probing (LP)

**1. Linear Probing (LP):** We trained a new linear head on top of the frozen backbone using the Waterbirds training set.

- **Duration:** 30 Epochs.

- **Learning Rate:** $5 \times 10^{-4}$.

- **Optimizer:** AdamW.

- **Batch Size:** 256.

- **Loss:** Weighted Cross-Entropy (to handle class imbalance).

## I.3. Baselines Implementation Details

**Data-Matched ERM Control** : this serves as a direct baseline to isolate the impact of our proposed alignment loss. This control utilized the exact same synthetic composites, augmentation pipeline, "virtual epoch" size and number of epochs as BAP, ensuring that any performance gains are not simply attributable to the data distribution or the synthesis process. Specifically, the experimental setup is identical to BAP in every respect—including the choice of backbone and the composition of training samples—with the sole exception of the objective function. Instead of minimizing the Cosine Embedding Loss against a robust anchor, the control was trained using standard Cross-Entropy loss on the ground-truth class labels. For this baseline, we employed a learning rate of $5 \times 10^{-6}$ for the backbone and the same learning rate schedulers as for the BAP deployment to ensure stable convergence under the supervised objective.

**Standard Linear Probe.**   We evaluate performance using a standard Linear Probe (LP) on the original LAION pre-trained weights. This protocol follows the identical evaluation setup used for both BAP and the Data-Matched ERM Control: the pre-trained visual backbone is frozen, and a linear classification head is optimized on top of the fixed feature representations. By using the same optimizer settings and hyperparameters, we ensure that any performance differences are driven solely by the quality of the learned representations.

**Linear Probe followed by Fine-Tuning (LP-FT).**   We employ a two-stage optimization strategy to ensure stability during downstream adaptation. In the first stage, we freeze the visual backbone and optimize only the linear classification head for 10 epochs to align the decision boundary. Subsequently, we unfreeze the encoder and fine-tune the entire model for an additional 40 epochs. This procedure utilizes the same optimizer configuration (AdamW) and learning rate schedules (linear warmup followed by cosine decay) described previously, maintaining a backbone learning rate of $5 \times 10^{-6}$ during the fine-tuning phase to prevent catastrophic forgetting.

**ERM (LP-FT) with RandAugment.**   To evaluate performance under diverse data augmentation, we employed the Linear Probe followed by Fine-Tuning (LP-FT) protocol with RandAugment applied to the training set. While keeping the optimization schedule identical to the standard LP-FT approach (10 epochs of linear probing followed by 40 epochs of fine-tuning), we introduced a modest RandAugment policy (Cubuk et al., 2020). This configuration applied $N = 3$ sequential augmentation transformations with a magnitude strength of $M = 2$, ensuring the model was exposed to geometric and photometric variations without aggressive semantic distortion.

**Zero-Shot:**   For zero-shot evaluation, we construct a robust textual classifier by ensembling multiple prompt templates. Instead of relying on a single prompt (e.g., "a photo of a {label}"), we utilize a standard set of diverse templates (such as "a bad photo of a {label}" or "a close-up photo of the {label}") to generate class-specific text embeddings. For each class $c$, we populate these templates with the class name and encode them using the pre-trained text encoder. The resulting embeddings are $L_2$-normalized, averaged to form a single prototype vector $\mathbf{w}_c$ for that class, and then re-normalized. Classification is performed by computing the cosine similarity between the image embedding $\mathbf{z}_i$ and each class prototype $\mathbf{w}_c$, assigning the label corresponding to the maximum similarity score. This ensembling strategy mitigates sensitivity to specific wording and generally improves zero-shot robustness.

**RoboShot**   We adapted the RoboShot framework (Adila et al., 2023) to serve as a zero-shot baseline for removing spurious correlations. While the original formulation employs Large Language Models (LLMs) to automatically discover potential nuisance attributes, we modified this protocol by bypassing the discovery phase. Instead, we leveraged our *a priori* knowledge of the dataset structure to explicitly define the nuisance variable set $\mathcal{Z}$ as the known background categories (e.g., *grass*, *water*, *land*).

Operatively, we first computed the standard zero-shot probability distributions for the target objects $P(Y|X)$ and the background attributes $P(Z|X)$ using the respective text prompt ensembles. We then utilized the RoboShot solver to effectively "subtract" the influence of the background features $Z$ from the prediction, solving the underlying linear system to estimate the debiased class probabilities. This allows us to evaluate the best-case performance of analytical debiasing when the spurious features are perfectly identified.

**External Baselines.**   For the comparative methods Wise-FT, AFR, DFR, and PruSC, we adhered strictly to the experimental setups and hyperparameter configurations detailed in their respective original publications (Wortsman et al., 2022; Qiu et al., 2023; Kirichenko et al., 2022; Le et al., 2024). This ensures that the reported results represent the intended performance of each method under its standard operating conditions.

### I.4. CounterAnimal Results

In this section, we detail the data processing, hyperparameter settings, and training procedures utilized for the CounterAnimal benchmark experiments. All experiments were conducted using the `open_clip` implementation of CLIP.

### I.5. Data Preprocessing and Augmentation

To ensure consistent evaluation across varying aspect ratios in the CounterAnimal dataset, we implemented a deterministic "Smart Crop" procedure rather than standard random resizing.

**Smart Crop Strategy:** For an input image with dimensions $(W, H)$:

- If $W > H$ (Landscape): We crop 50 pixels from both the left and right edges.

- If $H > W$ (Portrait): We crop 50 pixels from both the top and bottom edges.

- The resulting crop is resized to $224 \times 224$ using Bicubic interpolation with anti-aliasing enabled .

Standard CLIP normalization is applied to all images: $\mu = (0.481, 0.457, 0.408)$, $\sigma = (0.268, 0.261, 0.275)$. During training, we apply Random Horizontal Flipping. No other color or geometric augmentations were used to preserve the specific background correlations.

### I.5.1. BASELINE TRAINING CONFIGURATIONS

We compare our proposed BAP alignment method against a strictly data-matched ERM control. Both methods utilize the same synthetic composite generation pipeline (CUB objects composited onto Places365 backgrounds) to ensure that performance differences are attributable solely to the learning objective.

### I.5.2. BAP ALIGNMENT (PROPOSED)

The alignment phase minimizes the Cosine Embedding Loss between the student encoder's image embeddings and robust, foreground-averaged anchor vectors.

- **Dataset Size:** 12,000 synthetic composites (balanced between Landbirds and Waterbirds).

- **Anchor Generation:** Aggregated over $N = 10$ random backgrounds per object instance.

- **Optimizer:** AdamW with weight decay $\lambda = 0.01$.

- **Learning Rate:** $5 \times 10^{-6}$ (applied to the visual backbone).

- **Batch Size:** 256.

- **Training Duration:** 50 epochs.

- **Precision:** Mixed precision (bfloat16).

### I.5.3. DATA-MATCHED ERM CONTROL

The ERM control uses the identical synthetic dataset but is trained via standard Cross-Entropy loss on the class labels (Landbird vs. Waterbird) rather than embedding alignment. To ensure fair comparison and convergence, we utilize a two-stage training schedule:

- **Stage 1 (Linear Probe):** The backbone is frozen; the classification head is trained for 30 epochs with a learning rate of $1 \times 5^{-4}$.

- **Stage 2 (Fine-Tuning):** The backbone is unfrozen.

    - **Backbone LR:** $5 \times 10^{-6}$.
    - **Head LR:** $1 \times 5^{-4}$.
    - **Optimizer:** AdamW.
    - **Training Duration:** 40 epochs.

### I.5.4. Downstream Adaptation (Linear Probing)

After the pre-training (Alignment or Control) phase, we evaluate the robustified encoders on specific CounterAnimal pairs (e.g., Cheetah vs. Lion). We perform Linear Probing (LP) on the downstream task:

- **Data Split:** A 20% validation split is reserved from the training set.

- **Optimizer:** AdamW.

- **Learning Rate:** $1 \times 1^{-4}$.

- **Batch Size:** 32 (due to the small overall training set size).

- **Epochs:** 30.

- **Loss weighting:** To address class imbalance in specific pairs, we apply a weighted Cross-Entropy loss where the minority class weight is boosted by a factor of $\alpha = 2.0$ relative to the inverse frequency weight.

## I.6. NICO++ Pre-training Implementation Details

To evaluate the generalization of BAP beyond Vision Transformer (ViT) architectures and to scale to complex, real-world objects, we utilized the NICO++ benchmark. A critical distinction in this experimental phase is the transition from ViT-based backbones to a Convolutional Neural Network (CNN) architecture.

## I.7. Architecture and Pre-training Weights

For all NICO++ experiments, we employed the **ConvNeXt Base (Wide)** architecture (`convnext_base_w`) as implemented in the OpenCLIP library. For pre-training Weights, we utilized the `laion2b_s13b_b82k` checkpoint.

### I.7.1. Data Generation: COCO Vehicles

We constructed a synthetic training set by extracting foreground objects from the MS-COCO 2017 training split and compositing them onto backgrounds sampled from Places365.

- Target Classes: We filtered for six vehicle classes: *car, bus, airplane, boat, train,* and *bicycle*.

- Resolution Constraints: To ensure discriminative feature availability, we excluded object instances with bounding boxes smaller than $28 \times 28$ pixels.

## I.8. Hyperparameters and Training Schedule

We utilized a unified training duration of 160 epochs for both the alignment and control phases to ensure convergence.

### I.8.1. BAP Alignment

- Anchor Generation: Anchors were distilled using the `laion2b` teacher weights, aggregating embeddings over $N = 10$ random backgrounds per object instance.

- Learning Rate: $5 \times 10^{-6}$ for the visual backbone.

- Optimizer: AdamW with weight decay $\lambda = 0.01$.

- Scheduler: Linear warmup (400 steps) followed by Cosine Annealing decay.

- Batch Size: 512

- GPU: 1 H100

### I.8.2. DATA-MATCHED ERM CONTROL

The control baseline used the identical COCO+Places365 data stream but was optimized via Cross-Entropy loss.

- **Phase 1 (Linear Probe):** 5 epochs, Head LR $= 1 \times 10^{-3}$, Frozen Backbone.

- **Phase 2 (Fine-Tuning):** 160 epochs, Backbone LR $= 5 \times 10^{-6}$, Head LR $= 5 \times 10^{-4}$.

Evaluation on NICO++ follows the same linear probe and finetuning evaluation routine used for the Waterbirds and CounterAnimal sections.

