# OpenReview forum: "Birds of a Feather, Flocked Together? Deriving Background-Invariant Classifiers from CLIP Image Backbones"
_ICML.cc/2026/Conference — Submitted to ICML 2026_

### Official Review · Reviewer_RtDz · 2026-03-06

**Soundness:** 4
**Presentation:** 3
**Significance:** 3
**Originality:** 3
**Overall Recommendation:** 3
**Confidence:** 4

**Summary:**

This paper experimentally identifies the near-linear separability of foreground and background within the CLIP embedding space and proposes training CLIP with foreground-only anchor vectors to leverage this property. The proposed method is annotation-free and achieves promising results on Waterbirds, CounterAnimal, and NICO++.

**Compliance With Llm Reviewing Policy:**

Affirmed.

**Final Justification:**

Thank you for the authors’ response. After carefully reading the rebuttal and the comments from other reviewers, most of my concerns have been addressed. Nevertheless, in light of the relevant points raised by other reviewers, I currently hold a neutral stance on this manuscript. While the findings of this work (e.g., the linear superposition hypothesis) represent a valuable contribution, there remain certain issues regarding the theoretical proofs, writing quality, and experimental design. Although the authors’ rebuttal has satisfactorily addressed most of the reviewers’ concerns, incorporating these clarifications into the manuscript would require extensive revisions. Therefore, I have decided to keep my final overall rating unchanged, but I will increase the Significance score to 3 and the Soundness score to 4.

**Key Questions For Authors:**

The importance of weaknesses is ranked as: 2 > 1 $\approx$ 3

**Limitations:**

yes

**Strengths And Weaknesses:**

## **Strengths**
1. The method proposed in this paper is simple, intuitive, and easy to reproduce.
2. The hypothesis regarding the near-linear separability of foreground and background within the CLIP embedding space identified in Section 4 is highly valuable.
3. The performance of BAP appears to be strong.
4. The authors acknowledge the limitations of BAP in Section 7, which enhances the reliability of this paper.
## **Weaknesses**
1. Multiple studies from the community over the past year have explicitly pointed out that CLIP is very fragile when handling foreground-background decoupling. Relevant works include [1] and [2]. Therefore, repeatedly demonstrating that CLIP suffers from High Background Bias seems unnecessary, which diminishes the contribution of the experiments in Section 3.
2. The experimental evidence for the near-linear separability of foreground and background representations in the CLIP embedding space appears unreliable. Is this phenomenon solely due to the use of a synthetic dataset? The distinction between foreground and background in such data is extremely large, allowing the model to quickly identify these unnatural patterns. Does this linear separation also hold for foregrounds and backgrounds in real-world images?
3. BAP generates a set of synthetic images by superimposing the object onto randomly sampled background images, then derives $\bar{z}$ by calculating the average feature embedding of all images for training. If the authors' hypothesis in Section 4 holds (i.e., $v_{a,b} \approx v_a + v_b$​), does this imply that $\bar{z}$  contains not only an average foreground feature vector but also noise composed of the average of $K$ different background features? Would this not impact the training in the second stage?


[1] Wang, Qizhou, et al. "A sober look at the robustness of clips to spurious features." *Advances in Neural Information Processing Systems* 37 (2024): 122484-122523.

[2] You, Chenyu, et al. "Uncovering memorization effect in the presence of spurious correlations." *Nature Communications* 16.1 (2025): 5424.

---

> ### Author Rebuttal · Authors · 2026-03-29
>
> We thank the reviewer for their thoughtful feedback and the time invested in evaluating our work.
>
> * **Regarding W2: Is linear separability an artifact of synthetic data, and does it hold for real-world images? (Rank 1 Concern)**
>   * The reviewer presents a highly intuitive hypothesis: that cut-and-paste artifacts artificially inflate the additivity score.
>   * However, **our supervised controls in Section 4 provide evidence against this explanation**. We evaluated supervised IN1K models on the exact same synthetic composites. If additivity were purely driven by visual artifacts, IN1K models would also show similarly high additivity but they don't.
>   * **Text Encoder Evidence:** Additionally, Appendix A (Table 5) shows that CLIP's text encoder exhibits the same near-linear separability when combining pure text strings. Text strings contain zero visual artifacts, confirming this is an intrinsic multimodal feature of CLIP's representation space.
>   * **Real-World Transfer:** Finally, the fact that BAP robustness transfers to real-world datasets like NICO++ and CounterAnimal (Tables 3 & 4) proves (albeit indirectly) that this linear separability holds and can be exploited in real images.
>
> * **Regarding W1: The necessity of Section 3 given existing literature on CLIP's background bias.**
>   * We entirely agree that CLIP's background bias is already established. We explicitly cite Wang et al. (2024) and utilize their CounterAnimals benchmark.
>   * **Rare Spurious Regime:** However, prior works primarily focus on high correlation settings. Our core contribution in Section 3 is quantifying CLIP's behavior under the rare spurious regime ($\alpha \le 0.1$).
>   * We demonstrate that **CLIP retains catastrophic errors (~50%) even when the spurious cue appears in only 10% of the training data**. This finding justifies our aggressive representation-level intervention.
>   * We are happy to modify the language to highlight that our contribution lies in quantifying rare correlations rather than discovering CLIP's fragile foreground-background decoupling.
>
> * **Regarding W3: Does averaging backgrounds leave a residual noise vector that impacts training?**
>   * Theoretically, averaging K composites does indeed leave a residual background vector. However, as $K$ increases across randomly sampled diverse backgrounds, this residual converges toward a constant, class-agnostic offset rather than zero.
>   * **Non-Discriminative Residual:** Because this residual vector becomes uniform across all generated anchors, it carries no discriminative spurious information and therefore does not negatively impact downstream training.
>   * **Empirical Evidence:** Figure 5 (App. E) supports this: as the number of background composites averaged increases, similarity to the true foreground-only text embeddings consistently rises. This proves that coherent, image-specific background noise is successfully attenuated, isolating the foreground signal.
>   * **Functional Validation:** Finally, Table 6 confirms this functionally: the representational shift caused by background swaps is an order of magnitude lower with BAP than native CLIP. This would be highly unlikely if the anchors retained discriminative or variable background signals.
>
> We hope that the above comments address the concerns raised. We look forward to receiving your response and would be happy to clarify further on any points.

---

> > ### Author Rebuttal · Reviewer_RtDz · 2026-04-02
> >
> > - Regarding the response to W1, I am particularly interested in the observation that *"CLIP retains catastrophic errors (~50%) even when the spurious cue appears in only 10% of the training data."* Could the authors further elaborate on why this phenomenon occurs? This aspect does not appear to be thoroughly discussed in Sec. 3.1 of the original paper. Additionally, could you explain why the method proposed in this work is able to address this issue?
> >
> > - Regarding the statement in the response to W3 that *"this residual converges toward a constant, class-agnostic offset rather than zero,"* could you provide a formal proof or theoretical justification for this claim? Does this also imply that the proposed method is highly sensitive to the hyperparameter $K$ in Equation 2 of the original paper? It seems evident that when $K$ is small, the residual cannot converge properly. This interpretation appears to be corroborated by the results in Table 9 of the original paper.

---

> > > ### Author Response · Authors · 2026-04-02
> > >
> > > These follow-up questions highlight important nuances, which we are glad to elaborate on below.
> > >
> > > **Regarding CLIP's catastrophic errors (~50%) even with a spurious cue rate of 10%**
> > >
> > > Several factors account for this phenomenon, they include:
> > >
> > > * **Dataset Design:** Traditional benchmarks like Waterbirds utilize counterfactual foreground-background swaps (e.g., landbirds on water) as minority groups. The presence of these counterfactual swaps penalizes models for relying on spurious cues. In our rare spurious setup, however, we omit these counterfactuals. As a result, the 10% background acts as a perfectly predictive shortcut with zero penalty. This setup mimics highly realistic deployments since real-world data rarely provides convenient, symmetric counterfactuals. We invite the reviewer to view Section 3, Lines 111-150 for further reading on our dataset construction.
> > >
> > > * **Significance of Counterfactual BG-FG Swaps:** Table 2 demonstrates the impact these counterfactual swaps have: when minority groups are removed in the Waterbirds-100% setting, CLIP linear probe performance collapses from 61.6% to 24.5% WGA.
> > >
> > > * **CLIP's Specific Vulnerability:** Further to our dataset design, CLIP's vulnerability stems directly from the compositional 'bag-of-features' nature of the embedding space (Section 4). Empirically, **the error rate rankings in Figure 2 exactly match the linear additivity rankings in Table 1** (CLIP-Convnext $\approx$ CLIP-ViT $>$ IN1K-ViT $>$ IN1K-Convnext), with the most robust model being the one with the lowest linear additivity. This confirms that higher feature additivity increases reliance on spurious cues. Mechanistically, high additivity encodes the background as an isolated vector that a downstream linear probe can easily exploit as an independent shortcut.
> > >
> > > * **How BAP Mitigates the Issue:** BAP resolves this via an **information bottleneck**. By mapping multiple background composites to a single anchor vector, the model is mathematically forced to discard high-variance, unshared features (backgrounds) and retain only shared, core features (the foreground). Our Random Orthogonal Target ablation (Appendix C) confirms this; we replaced the foreground anchors with two random orthogonal vectors and mapped all waterbird composites to one vector and all landbird composites to the other using the same cosine similarity loss. Results in Table 7 show strong performance is retained on the Waterbirds benchmark, which validates our theory on BAP working as an information bottleneck. We note that OOD performance suffers significantly under this ablation indicating that the teacher-generated semantic anchors are essential.
> > >
> > > ***
> > >
> > > **Regarding Anchor Vector Residuals**
> > >
> > > We appreciate the opportunity to clarify the theoretical framework of our anchor generation phase using the properties of expected values and the Law of Large Numbers.
> > >
> > > * **1. Convergence Definition:** Based on Section 4, a composite embedding $z_{k}$ is approximately the linear superposition of the foreground and background vectors **(up to an $L_2$ normalization scalar)**:
> > >     $$z_{k} \approx v_{fg} + v_{bg_{k}}$$
> > >     In Phase 1, our anchor $a$ averages $K$ diverse composites from a broad distribution (e.g., Places365):
> > >     $$a \approx v_{fg} + \frac{1}{K} \sum_{k=1}^{K} v_{bg_{k}}$$
> > >     By the Law of Large Numbers, as $K$ increases, the sample mean of the backgrounds converges to its expected value: $$\mu_{bg} = \mathbb{E}[v_{bg}]$$
> > > Because backgrounds are sampled randomly and independently of the foreground, $\mu_{bg}$ is strictly **class-agnostic** and contains no discriminative information.
> > >
> > > * **2. A Non-Zero Offset:** CLIP embeddings exist on a high-dimensional unit hypersphere, but natural images only occupy a narrow cone within it [1]. Consequently, the centroid of all backgrounds ($\mu_{bg}$) is a strictly non-zero vector pointing toward this general image manifold.
> > >
> > > * **3. Sensitivity to $K$ and Residual Noise:** For a finite $K$, the anchor retains residual noise, defined as:
> > >     $$\epsilon = \left( \frac{1}{K} \sum_{k=1}^{K} v_{bg_{k}} \right) - \mu_{bg}$$
> > >     Assuming the random backgrounds are drawn independently (such that i.i.d holds), the variance of this residual noise scales inversely with $K$:
> > >     $$Var(\epsilon) \propto \frac{1}{K}$$
> > >     This explains the asymptotic curve in Figure 5 and the results in Table 9. The noise variance drops sharply at small values of $K$ (e.g., 1 to 4), producing rapid robustness gains. However, as $K$ increases, the marginal variance reduction shrinks, and the background term stabilizes at the constant offset $\mu_{bg}$. Therefore, while BAP requires $K > 1$ to suppress noise, it is not highly sensitive at large $K$ values.
> > >
> > > * *Note: We will add this explanation to the Appendix as supplementary theoretical material.*
> > >
> > > [1] : https://arxiv.org/pdf/2504.11695
> > >
> > > We thank the reviewer for their high level of engagement with our work and hope the above comments address the concerns raised.

---

### Official Review · Reviewer_FxuC · 2026-03-12

**Soundness:** 3
**Presentation:** 2
**Significance:** 3
**Originality:** 2
**Overall Recommendation:** 3
**Confidence:** 4

**Summary:**

This paper analyzes the background bias problem where CLIP-based image encoders excessively rely on background information and proposes a method to mitigate it. The authors argue that CLIP encoders exhibit "bag-of-features" characteristics, representing scenes as a linear superposition of foreground and background, which makes them vulnerable to spurious correlations. To address this, the authors propose a two-stage pre-training method called BAP (Background-Averaged Pretraining). First, they generate images by synthesizing various backgrounds while keeping the same foreground object. These images are fed into a CLIP encoder, and the resulting embeddings are averaged to construct a foreground-invariant representation. Finally, a student encoder is trained to align with this representation, learning features that are less dependent on the background. Evaluating a simple classifier on top of the learned encoder, the authors report a Worst-Group Accuracy (WGA) of over 90% on the Waterbirds benchmark under a 100% spurious correlation setting, outperforming existing methods. Furthermore, to evaluate generalization, experiments were conducted on the CounterAnimal OOD dataset, supporting the claim that mitigating background bias improves overall generalization performance.

**Compliance With Llm Reviewing Policy:**

Affirmed.

**Final Justification:**

I thank the authors for their rebuttal. However, I believe the core idea remains relatively simple and lacks sufficient novelty compared to existing approaches. For this reason, I agree with other reviewers regarding the limited originality of the work.

**Key Questions For Authors:**

- The proposed method relies on the assumption that averaging embeddings of images with the same foreground but different backgrounds yields a foreground-invariant representation. Is there any empirical evidence or experimental validation confirming that this averaging process effectively suppresses background noise while preserving critical foreground information?

- The pipeline appears heavily dependent on the process of isolating foreground objects and compositing them with new backgrounds. How robust is the proposed method when foreground masks are of low quality or when segmentation is imperfect?

- Is there a possibility that this compositing-based approach is specifically optimized for datasets with relatively simple structures like Waterbirds? Do you have further analysis on whether the same effects hold for datasets with greater foreground variation or scenarios where foreground masks are ambiguous?

**Limitations:**

- The reliance on foreground segmentation or mask extraction may significantly limit the practical applicability of the proposed method in real-world environments.

- Images generated via background compositing may deviate from natural distributions, potentially exerting unintended influences on representation learning.

- While the approach may be effective on spurious correlation benchmarks like Waterbirds, it remains unclear whether these effects hold for general datasets characterized by high foreground variation.

- Recent studies report that Worst-Group Accuracy (WGA) on the Waterbirds benchmark has already reached the 85–95% range. Consequently, performance gains on this specific benchmark may not necessarily signify a fundamental methodological breakthrough.

**Strengths And Weaknesses:**

Strengths

- The proposed method features a relatively straightforward architecture utilizing foreground-background compositing and embedding averaging. It offers an intuitive approach that directly breaks the background shortcut without excessive implementation complexity.
- While existing research on spurious correlation primarily focuses on sample reweighting or group optimization, this study distinguishes itself by attempting to make the representation itself background-invariant.
- By conducting experiments on the CounterAnimal dataset in addition to the Waterbirds benchmark, the authors demonstrate that their method is not merely overfitted to a specific benchmark.
- The method is implemented as a teacher-student distillation framework, providing a structure that can be easily applied to existing CLIP encoders.

Weaknesses
- The paper presents the idea that CLIP encoders represent scenes as a linear combination of foreground and background as a major discovery; however, similar observations have been repeatedly reported in recent studies. It is difficult to conclude that this paper provides a systematic proof or a truly original analysis of this phenomenon.
- Some of the listed contributions appear to be restatements of phenomena already widely discussed in previous literature rather than novel technical contributions. Specifically, the explanation of "bag-of-features representation" can hardly be viewed as a new finding.
- The proposed method presupposes foreground segmentation and background compositing. While feasible for specific datasets like Waterbirds, this approach may be difficult to generalize to complex, natural image datasets.
- The assumption that averaging embeddings with randomized backgrounds will recover the foreground representation is intuitively plausible, yet the paper lacks clear evidence or a theoretical basis proving that CLIP representations possess such an additive structure.
- The claim that the student encoder provides meaningful improvements over the original CLIP representation relies heavily on comparison results with a CLIP linear probe. If the persuasiveness of this comparison is not sufficiently emphasized, the perceived contribution may be limited.

---

> ### Author Rebuttal · Authors · 2026-03-29
>
> We thank the reviewer for their feedback & the time invested.
>
> * **Concern: Does the averaging process suppress background noise while preserving foreground information?**
>   * We shared this concern initially; Appendix E & Figure 5 address this.
>   * **Foreground Preservation:** Figure 5 shows the cosine similarity between our generated anchors & foreground-only text prompts. As the number of composites averaged increases, similarity to the foreground-only text embedding consistently rises. This demonstration foreground information is preserved during the averaging process.
>   * **Background Suppression:** This rising similarity is consistent with attenuation of incoherent background noise. Additionally, Table 6 (Appendix B) shows the representational shift from background swaps is an order of magnitude lower with BAP than native CLIP, which would not very unlikely if the anchors contained coherent background signals.
>
> * **On the generalization to high foreground variation data & general real-world datasets.**
>     * **Real-world evaluation:** We evaluate BAP on two real-world benchmarks: CounterAnimal & NICO++ (Tables 3 & 4). NICO++ specifically tests domain generalization & contains high foreground variation & object occlusions. Tables 3 & 4 demonstrate strong robustness transfer to real-world data.
>   * **Super-Class Generalization:** To stress test generalization, the "truck" class was excluded from NICO++ pre-training data. Despite this, BAP achieved a **30% WGA improvement** over native CLIP on the "Car vs. Truck" task, proving generalization to the super-class level.
>
> * **Regarding sensitivity to segmentation quality & imperfect masks.**
>   * This reviewer raises a critical engineering overhead; we address this directly in Appendix D.
>   * Table 8 provides a sensitivity analysis of BAP across various levels of segmentation quality. Performance is consistent across mask quality: WGA drops less than 2% from perfect masks (91.8%) to "Noisy" (89.9%) or "Botched" (90.5%) masks. **Perfect segmentation is not required for near-optimal performance**.
>
> * **Concern: The reliance on segmentation or mask extraction limits practicality.**
>   *  This is another critical overhead; our analysis in Appendix F demonstrates it is a highly manageable prerequisite.
>   * **Low Data Requirement:** Figure 7 shows near-peak robustness (WGA > 90% on Waterbirds) is achieved using **as few as 100 segmented foreground items**. Combined with BAP's tolerance for coarse bounding boxes & the availability of tools such as SAM, extracting this initial set is highly practical.
>   * **One-Time Cost:** Finally, note that BAP is a one-time pre-training step. As demonstrated by our NICO++ & CounterAnimal evaluations, downstream linear probing & inference require no masks.
>
> * **On the evidence that CLIP representations possess an additive structure.**
>   * We kindly draw attention to Section 4 & Appendix A which demonstrate that CLIP has an additive structure in both image & text encoders respectively. This claim is further supported by recent works, such as SpLiCE (NeurIPS 2024), which we will reference in the camera ready version.
>
> * **Concern: That CLIP encoders represent scenes as linear combinations is not a major discovery.**
>   * We fully agree & will modify the language in the camera-ready version to reflect this.
>   * However, we introduce a quantitative metric ($Sim_{Both,Sum}$) which provides a novel, intuitive formulation for quantifying foreground-background entanglement. Furthermore, to our knowledge, we are the first to leverage this specific insight as the direct foundation for a novel robustness intervention.
>
> * **Concern: Improvements over the original CLIP representation rely on comparison with a CLIP linear probe.**
>   * Indeed, a direct comparison between BAP & original CLIP is unfair due to BAP's increased data exposure.
>   * To isolate BAP's algorithmic contribution from mere increased data exposure, we introduced (line 273) the "data-matched control" (LP-FT using the same synthetic data used by BAP). Across all results, BAP exhibits higher WGA compared to the data-matched control proving improvements are not solely due to increased data exposure.
>
> * **Concern: Waterbirds is a saturated benchmark.**
>   * We agree Waterbirds-95% is saturated. However, Waterbirds-100% & by extension 100% spurious correlation in general, remains a difficult, unsolved problem.
>   * Traditional methods (DFR, AFR, JTT, CnC) break down in such regimes due to the lack of minority group data [1] while BAP is the first method to maintain high performance in the 100% regime, achieving a 91.8% WGA (Table 2). This represents a substantial **12% improvement over the current state-of-the-art** [1].
>  * Finally, note the results Tables 3 and 4 were obtained with 100% spurious correlation setups.
>
> [1] : https://neurips.cc/virtual/2025/loc/san-diego/127227
>
> We look forward to receiving your response and would be happy to clarify further on any points.

---

> > ### Author Rebuttal · Reviewer_FxuC · 2026-04-03
> >
> > The authors' responses are generally sincere and valid. However, the key limitations raised in this review (the generalizability of the method, strong assumptions, and particularly limited originality) appear to remain valid, and accordingly, I will not change my final assessment.

---

> > > ### Author Response · Authors · 2026-04-03
> > >
> > > We sincerely appreciate the reviewer’s continued engagement during the discussion phase and appreciate the acknowledgement that the rebuttal responses were 'sincere and valid.'
> > >
> > > While we respectfully maintain that the out-of-distribution evaluations on CounterAnimal and NICO++ empirically address theoretical concerns regarding generalizability, the reviewer's perspective on the underlying assumptions and the framing of originality is highly valued. We will ensure that these specific nuances, as well as the inherent limitations of compositing, are discussed even more thoroughly in the final version of the manuscript.
> > >
> > > We thank the reviewer once more for the time invested, the constructive feedback, and the support of the paper with a positive score.

---

### Official Review · Reviewer_4XRS · 2026-03-16

**Soundness:** 3
**Presentation:** 3
**Significance:** 2
**Originality:** 2
**Overall Recommendation:** 3
**Confidence:** 4

**Summary:**

This paper studies spurious correlations in CLIP models between the background and foreground objects. The paper proposes a finetuning method to make the model group-robust that is based on synthetic data that mimics foreground background decorrelation. Results are shown on the classic water/landbird, CounterAnimal dataset and vehicle classification.

**Compliance With Llm Reviewing Policy:**

Affirmed.

**Final Justification:**

I thank the authors for the rebuttal. While the point about DINO model family was cleared, the point remains about SigLIP. Furthermore, while the authors made clear that some researchers in the robustness community do not mind  visual representation qualities suffering when increasing robustness, I, as a reviewer in the general vision/ML community disagree and see this as a weakness, as a very strong pretrained backbone is effectively weakened.

**Key Questions For Authors:**

* some tables are with ConvNext, some with ViT? why are not both results shown?
* after finetuning, classic LP measurements on the standard CV datasets such as IN1k, CUB/Aircraft/flowers/places etc. would be useful to determine if the improved foreground-background separation performance comes at a cost of general feature quality.
* if the latter, can this be mitigated by finetuning with PEFT such as LoRA or VeRA?

**Limitations:**

yes

**Strengths And Weaknesses:**

Strength:
*evaluates not only CLIP but also an IN1k trained model and ConvNeXt architecture.
* background-foreground bag-of-wordsness is nicely quantified in a simple manner using the cosine similarities paired object/background sets of images.
* relatively simple training method


Weaknesses:
* focussing solely on CLIP: what about SigLIP(v2) and other vision models? the text-component is not essential so extending the analysis to further sota backbones would be important (DINOv2/3, MAE etc.).
* the analysis in section 4 ignores the magnitude of the resulting embeddings, these might carry some further information.
* some tables are with ConvNext, some with ViT and are not consistent in terms of backbone.
* different sizes of backbones are not analysed and whether these biases persist in bigger and importantly densely pretrained (SigLIPv2, DINOv2) backbones.
* no analysis on the created synthetic data: creation hyperparameters and dataset size (only parameter K is ablated in the appendix)
* it’s unclear if the finetuned backbone gets worse in its visual representation.
* GradCAM results are highly cherry-pickable and not indicative of actual processing in these models, see e.g. “Sanity Checks for Saliency Maps” NeurIPS 2018

Other:
Formatting not correct in the pdf: line 314 shows a negative vspacing
line 1329: learning rate of “1^{-4} “is  likely a typo. powers of 1 .. remain 1

---

> ### Author Rebuttal · Authors · 2026-03-28
>
> We thank the reviewer for their thoughtful feedback and the time invested in evaluating our work.
>
> * **Evaluating vision-only models (e.g., DINO) and exhaustive VLM benchmarking is distinct from our targeted theoretical framework.**
>   * **CLIP-Specific Formulation:** We demonstrate empirically that CLIP models contain a high degree of linear additivity between foreground & background representations in both image & text encoders (Section 4 & Appendix A respectively). BAP is formulated to explicitly exploit this specific property inherent to CLIP representations.
>   * **Scope of Generalization:**  While an exciting prospect, extending BAP to vision-only models without text-supervision requires distinct theoretical formulations, since we observed lower linear additivity for supervised IN1K models (Table 1).  Newer VLMs like SigLIP share the same pre-training paradigm as CLIP and would likely benefit; however, exhaustive benchmarking across all modern VLM architectures exceeds the scope of this targeted mitigation.
>
> * **Comprehensive ablations on dataset size & creation hyperparameters are fully detailed in Appendix F.**
>   * We appreciate the formatting notes & will correct the line 314 vspace & broken equation reference.
>   * **Ablation Coverage:** Regarding the K parameter ablation (App. F.0.2); App. F.0.1 shows our ablation on dataset size requirements while an ablation on the number of background composites required is shown in App. F.0.3. We will reference the corresponding figures more prominently in the main text to improve visibility of these analyses as this could have been clearer in the original submission.
>
> * **Regarding embedding magnitudes carrying additional information.**
>   * **Measuring Compositional Additivity via Angular Alignment:** Our objective in Section 4 is strictly to quantify compositional additivity, which is standardly evaluated via angular alignment (cosine similarity). CLIP representations are unit-normalized (i.e., lie on a hypersphere) (Section 4, line 148).
>   * **Normalization Clarification:** To ensure clarity, the camera-ready version will explicitly state that supervised encoder representations were also unit-normalized for a fair geometric comparison. Supervised embedding magnitudes (pre-normalization) typically encode feature activation strength, not the compositional structure we are measuring.
>
> * **Employing both ViT & ConvNeXt was designed to demonstrate cross-architectural utility.**
>   * **BAP is Architecture-Agnostic:** We deliberately varied the backbone to demonstrate BAP’s robustness gains are architecture-agnostic. For direct comparability, we will add ConvNeXt Waterbirds results to the appendix.
>   * **Consistent Performance:** ConvNeXt performs very similarly to ViT on Waterbirds-100% (91.5% WGA for BAP vs. 78.3% for the ERM control, with all other baselines having lower WGA), confirming consistent gains regardless of the visual backbone.
>
> * **The cost to general feature quality is an intended mechanical outcome, quantified in Appendix G.**
>   *  Indeed, the reviewer correctly notes that general scene recognition degrades; we quantify this "context-blindness" in Appendix G. **The core objective of BAP is to intentionally decouple background associations from foreground representations** (a strict requirement in safety-critical domains like medical imaging).
>   * **PEFT Usage:** Consequently, applying PEFT methods (e.g., LoRA) to retain scene recognition while simultaneously inducing background invariance would most likely counteract the method's goal of isolating the foreground target.
>
> * **Grad-CAM is utilized strictly for qualitative illustration, distinct from our quantitative claims.**
>   *  We agree with the reviewer regarding the limitations of saliency maps & do not rely on them to validate BAP's efficacy.
>   * **Quantitative Claims:** Our core conclusions & claims rest entirely on quantitative benchmarking, ablations, & hyperparameter sweeps across Tables 1 through 10 & Figures 2, 5, 6, 7, & 8.
>
> We hope that these comments address the concerns raised. We look forward to receiving your response & would be happy to clarify further on any points.

---

> > ### Author Rebuttal · Reviewer_4XRS · 2026-04-03
> >
> > I thank the authors for the rebuttal.
> > Two key issues raised in my review remain unaddressed, as also noted in the rebuttal: 1) the extension to more than just the CLIP model (very important, since the community has moved to SigLIPv2 or DINOv2 models). 2) The visual representation qualities suffer under this approach, so the resulting model cannot be used as a general strong pretrained backbone.

---

> > > ### Author Response · Authors · 2026-04-03
> > >
> > > We thank the reviewer for summarizing their remaining concerns. We respectfully offer the following brief clarifications to contextualize the remaining points:
> > >
> > > * **1. Architectural Scope (e.g., DINOv2):** BAP explicitly leverages the linear additivity of a joint text-image embedding space which our work shows is specific to CLIP. While exploring background-invariance in self-distilled, vision-only paradigms is an exciting future direction, it requires a fundamentally different theoretical formulation outside the scope of this paper.
> > >
> > > * **2. The Continued Relevance of CLIP:** While we acknowledge the rapid evolution of foundation models, CLIP remains a critical backbone for modern Large Vision-Language Models. Mitigating its specific spurious correlations remains a highly active area of research; **multiple works from late 2025 address CLIP's vulnerabilities at top venues** (e.g., SAGE [1], COLA [2], LEAF [3]). Establishing strict background-invariance for CLIP architectures remains an important, unsolved problem.
> > >
> > > * **3. Context-Blindness as an Intended Trade-off:** We completely agree with the reviewer that BAP degrades general visual context (Appendix G), making it unsuitable as a general-purpose backbone. However, the robustness community has established a fundamental tension between standard accuracy (relying on all context) and robust accuracy (discarding spurious features) [4]. In safety-critical domains like medical imaging—where models infamously cheat using background scanner artifacts [5]—retaining general context introduces severe vulnerabilities. BAP deliberately trades general scene recognition for guaranteed safety in these specific, high-stakes settings.
> > >
> > > We appreciate the reviewer's time and believe our methodology, scope, and trade-offs are well-supported by recent literature.
> > >
> > > To ensure clarity for future readers, we will explicitly highlight this intentional trade-off regarding context-blindness, as well as the specific architectural dependencies of BAP, in the Limitations and Discussion sections of the camera-ready manuscript.
> > >
> > > [1] https://arxiv.org/abs/2511.13005 | Accepted at AAAI 2026.
> > >
> > > [2] https://neurips.cc/virtual/2025/loc/san-diego/poster/115629 | *NeurIPS* 2025.
> > >
> > > [3] https://neurips.cc/virtual/2025/loc/san-diego/poster/118180 |  *NeurIPS* 2025.
> > >
> > > [4] Tsipras et al., "Robustness May Be at Odds with Accuracy," *ICLR*, 2019.
> > >
> > > [5] Zech et al., "Variable generalization performance of a deep learning model to detect pneumonia in chest radiographs," *PLOS Medicine*, 2018.

---

### Official Review · Reviewer_HNmb · 2026-03-16

**Soundness:** 2
**Presentation:** 2
**Significance:** 2
**Originality:** 3
**Overall Recommendation:** 3
**Confidence:** 4

**Summary:**

This paper introduces Background-invariant Anchor Pre-training (BAP), a method designed to mitigate CLIP's reliance on background bias by leveraging the insight that CLIP encoders function as compositional "bag-of-features" models. By quantifying the tendency of CLIP to represent scenes as a linear superposition of foreground and background, the authors develop a two-phase pre-training protocol that aligns image representations with synthetic, foreground-only anchor vectors. BAP achieves state-of-the-art results, including over 90% Worst-Group Accuracy (WGA) on the Waterbirds benchmark even under perfect spurious correlation, demonstrating a robust, task-agnostic backbone that generalizes to unseen objects and facilitates strong sim-to-real transfer.

**Compliance With Llm Reviewing Policy:**

Affirmed.

**Final Justification:**

Thank the author for the rebuttal. However, the key limitations and questions (the generalizability of the method, strong assumptions) appear to remain valid, and some concerns have not been addressed well (the limited scope). I will not change my final assessment and tend to reject.

**Key Questions For Authors:**

1. Given the lack of lighting consistency and physical plausibility in "cut-and-paste" synthesis, will the model overfit to boundary artifacts?
2. How effectively can a classifier trained on such synthetic distributions generalize to real-world images with complex natural background coupling?
3. Random composition ignores ecological validity, potentially harming tasks where context is a helpful prior?

**Limitations:**

yes

**Strengths And Weaknesses:**

Strengths:
- Robustness: Excels under 100% spurious correlation (e.g., Waterbirds WGA > 90%).
- Plug-and-Play: Requires no architectural changes to the frozen CLIP backbone.

Weaknesses:
- Method: CLIP's embedding space is highly non-linear; linear averaging may fail to cancel background noise and instead create "feature artifacts." Moreover, the alignment loss may inadvertently suppress critical foreground details (color/texture) that resemble background patterns, reducing classification accuracy.
- If the frozen CLIP teacher has pre-existing background shortcuts, the "anchors" will be inherently biased, leading the student to replicate those errors.
- How to prove the proposed pre-trained strategy work better than traditional methods for domain shift problems?
- Experiments: More real-world cases and data should be validated and discussed. The current scope is too limited.

---

> ### Author Rebuttal · Authors · 2026-03-28
>
> We thank the reviewer for their thoughtful feedback and the time invested in evaluating our work.
>
> * **Concern: Overfitting to boundary artifacts and generalizing from synthetic to real-world images. (Questions 1 & 2)**
>   * We fully agree that generalizing from synthetic composites to real-world data is a critical test for this method. To address this, we tested BAP on two real-world datasets: CounterAnimals and NICO++ (Tables 3 and 4, Section 6).
>   * **Real-World Generalization:** Specifically, NICO++ (CVPR 2023) is designed to test domain generalization under complex natural background couplings and object occlusions. Results show BAP maintains high performance on these real-world images, proving strong generalization without relying on synthetic "cut and paste" artifacts.
>
> * **Concern: The method trains the model to ignore context, which can harm tasks where ecological validity is a helpful prior. (Question 3)**
>   * An astute observation; indeed, BAP trains the model to ignore context (quantified in Appendix G), which we agree is not desirable for all applications. However, this property is by design since BAP's goal is to create a strictly background-invariant feature extractor.
>   * **High-Stakes Domains:** In critical domains, background cues may serve as semantic backdoors. For example, in medical imaging, models often cheat by using scanner artifacts or text markers as shortcuts, leading to real-world failures [1, 2]. BAP is engineered specifically to ensure robustness in these high-stakes domains.
>
> * **Concern: Linear averaging typically fails to cancel background noise in highly non-linear embedding spaces.**
>   * This is a reasonable intuition regarding non-linear spaces, and one we initially shared. However, Section 4 and Appendix A (Tables 1 and 5) demonstrate that **CLIP's image and text embedding spaces exhibit a high level of linear additivity.** This finding is further supported by recent peer-reviewed work, such as SpLiCE (NeurIPS 2024) [3].
>   * **Exploiting Additivity:** BAP directly exploits the near-linear separability of foreground and background representation. For a further explanation on linear averaging recovering the true foreground representation, we refer the reviewer to the next point.
>
> * **Concern: Alignment loss might suppress critical foreground details, and the frozen teacher might impart biased anchors.**
>   * We address this directly in Appendix E and Figure 5 by measuring the cosine similarity between our generated anchor vectors and the corresponding species-specific text prompts (e.g., "a photo of a duck").
>   * **Foreground Retention:** Figure 5 clearly demonstrates that as the number of random background composites used in anchor construction increases, the cosine similarity to the true foreground-only text embedding consistently increases. This proves that foreground features are successfully retained and amplified during anchor generation and the teacher's pre-existing background biases are bypassed.
>
> * **Concern: How the pre-training strategy compares against traditional methods for domain shift.**
>   * We agree that robust benchmarking against established techniques is vital. We tested a variety of traditional methods, including linear probing, full fine-tuning (LP-FT), and a strict "data-matched control" (ERM LP-FT on the exact same synthetic data used by BAP) to isolate BAP's algorithmic contribution from mere increased data exposure.
>   * **Algorithmic Necessity:** Across all 7 sets of results, BAP exhibits noticeably higher Worst-Group Accuracy (WGA) compared to the data-matched control, proving the necessity of the BAP algorithm over standard ERM.
>   * **Broad Baselines:** Furthermore, Table 2 benchmarks BAP against a wide envelope of baselines, including representation-level methods (PruSC), last-layer retraining (AFR, DFR), and CLIP-specific techniques (RoboShot, WISE-FT).
>
> We hope that the above comments address the concerns raised. We look forward to receiving your response and would be happy to clarify further on any points.
>
> [1] : https://journals.plos.org/plosmedicine/article?id=10.1371/journal.pmed.1002683
>
> [2] : https://pubmed.ncbi.nlm.nih.gov/33196064/
>
> [3] : https://neurips.cc/virtual/2024/poster/96446

---

> > ### Author Rebuttal · Reviewer_HNmb · 2026-04-06
> >
> > Thank the author for the rebuttal. However, the key limitations and questions (the generalizability of the method, strong assumptions) appear to remain valid, and some concerns have not been addressed well (the limited scope), I will not change my final assessment.

---

> > > ### Author Response · Authors · 2026-04-06
> > >
> > > We thank the reviewer for their final assessment.
> > >
> > > We note that the concluding remarks regarding "limited scope," "generalizability," and "strong assumptions" remain at a high level and do not appear to reflect the detailed clarifications provided in our rebuttal.
> > >
> > > We believe the key concerns raised during the initial review were addressed through targeted analyses and experiments, as outlined in the discussion thread.
> > >
> > > We defer to the Area Chair’s judgement in weighing these points.
> > >
> > > Thank you again for your time and consideration.

---

### Decision · Program_Chairs · 2026-04-30

**Decision:**

Reject

**Comment:**

This paper studies spurious correlations in CLIP models between the background and foreground objects. The paper proposes a two-phase pre-training protocol that aligns image representations with synthetic data that mimics foreground-background decorrelation.

Initially, the paper obtained 3 weak rejections and 1 weak acceptance. After the rebuttal and discussion, all reviewers agreed that the paper cannot be accepted to ICML.
- Rev. HNmb has concerns about the generalizability of the method to real images, as it is trained on synthetic ones, and the limited evaluation on real data.
- Rev. 4XRS has concerns about the evaluation of the proposed approach on SigCLIP, and the fact that the proposed reboust approach reduces the clean performance of the model.
- Rev. FxuC, after an initial weak accept score, found the concerns on generalizability and the limited novelty to be the main reasons to change to weak reject.
- Rev. RtDz considered the contributions of the paper relevant. However, they also found the other reviewers' issues to be important. In addition, while all their points were clarified, the new results and explanations would require a major editing of the manuscript.

Overall, while the paper has some merits (simple but effective approach), I agree with the reviewers that it is not clear how well the proposed approach could generalize to real data, especially considering the limited validation on real datasets. In addition, the proposed method seems limited in novelty, as most of the presented observations were already done in previous papers. Therefore, my recommendation is to reject this paper.